# Adaptive responses of marine diatoms to zinc scarcity and ecological implications

Riss M. Kellogg [1], Mark A. Moosburner[2,3], Natalie R. Cohen [4,5], Nicholas J. Hawco [6], Matthew R. McIlvin[4], Dawn M. Moran[4], Giacomo R. DiTullio [7], Adam V. Subhas[4], Andrew E. Allen [2,3] & Mak A. Saito [4✉]

Scarce dissolved surface ocean concentrations of the essential algal micronutrient zinc suggest that Zn may influence the growth of phytoplankton such as diatoms, which are major contributors to marine primary productivity. However, the specific mechanisms by which diatoms acclimate to Zn deficiency are poorly understood. Using global proteomic analysis, we identified two proteins (ZCRP-A/B, Zn/Co Responsive Protein A/B) among four diatom species that became abundant under Zn/Co limitation. Characterization using reverse genetic techniques and homology data suggests putative Zn/Co chaperone and membrane-bound transport complex component roles for ZCRP-A (a COG0523 domain protein) and ZCRP-B, respectively. Metaproteomic detection of ZCRPs along a Pacific Ocean transect revealed increased abundances at the surface (<200 m) where dZn and dCo were scarcest, implying Zn nutritional stress in marine algae is more prevalent than previously recognized. These results demonstrate multiple adaptive responses to Zn scarcity in marine diatoms that are deployed in low Zn regions of the Pacific Ocean.

[1] MIT/WHOI Joint Program in Oceanography/Applied Ocean Science and Engineering, Woods Hole, MA, USA. [2] Microbial & Environmental Genomics, J. Craig Venter Institute, La Jolla, CA, USA. [3] Scripps Institution of Oceanography, Integrative Oceanography Division, University of California, San Diego, La Jolla, CA, USA. [4] Department of Marine Chemistry and Geochemistry, Woods Hole Oceanographic Institution, Woods Hole, MA, USA. [5] Skidaway Institute of Oceanography, University of Georgia, Savannah, GA 31411, USA. [6] University of Hawai'i at Mānoa, Honolulu, HI 96822, USA. [7] Hollings Marine Laboratory, College of Charleston, Charleston, SC, USA. ✉email: msaito@whoi.edu

Marine diatoms are abundant photoautotrophic algae that contribute significantly to photosynthetic carbon fixation and export in the oceans through their participation in the biological carbon pump[1]. Among the micronutrients required for diatoms metabolism, zinc (Zn) is known to be particularly important[2]. However, dissolved Zn (dZn) can be extremely scarce in the upper ocean with dZn surface concentrations up to two orders of magnitude lower than those of the deep ocean due to uptake in the photic zone and regeneration at depth[3,4]. Surface depletion is due to the high biological demand for Zn within metalloenzymes such as carbonic anhydrase (CA), alkaline phosphatase, and proteases, which are essential to the processes of carbon fixation, organic phosphorus uptake, and remineralization, respectively[5,6]. As an adaptation to this scarcity, many marine diatoms are capable of metabolically substituting cobalt (Co) and cadmium (Cd) for Zn within certain subclasses of CA[2,7–10]. This substitution ability implicates metabolic systems in diatoms with the specific purpose of responding and adapting to low Zn, though proteins specific to these pathways have remained uncharacterized.

The role of Zn in influencing marine primary production is not as clear as that of iron (Fe), though frequent observations of surface open-ocean total dZn reaching only tens of picomoles per liter (0.041 ± 0.007 nmol/kg, GEOTRACES Zn surface water reference standard GS) suggests the potential for Zn to act as a primary or co-limiting nutrient[5,11,12]. However, the highly contamination-prone nature of Zn experimentation has compounded the difficulty in detecting Zn limitation in the field. Furthermore, while there is extensive work characterizing the bacterial response to Zn limitation[13–16], the cellular response to Zn stress at the proteome level in diatoms remains poorly understood.

In this study, we identified two Zn- and Co-responsive proteins in four distinct diatom species using a global proteomic approach and explored the abundance patterns of these proteins as a function of $Zn^{2+}$ and $Co^{2+}$ media concentrations. We employed reverse genetic techniques to further characterize these proteins and their putative functions. Coupling our laboratory findings to the field, we then present metaproteomic data demonstrating the presence of these proteins in surface Pacific waters. The presence of these proteins in the field implicates Zn stress within natural phytoplankton communities. Overall, this multidisciplinary study demonstrates that diatoms have a complex adaptive response to Zn scarcity, and that this scarcity may be a more important influence on primary productivity than previously recognized.

## Results and discussion

**Identification of two Zn/Co responsive proteins in diatoms**. Zn and Co growth rate experiments in which Zn or Co (omitting the other) were added to the growth media were conducted and harvested for proteomic analysis. Growth rates of the marine diatom species *Thalassiosira pseudonana* CCMP1335, *Phaeodactylum tricornutum* CCMP632, *Pseudo-nitzschia delicatissima* UNC1205 and *Chaetoceros sp.* RS19 (*Chaetoceros* RS19 herein) were conducted in a consistent media composition to allow for intercomparison among species (see "Methods"). The onset of growth limitation by Zn and Co was evident by decreased growth rates under low $[Zn^{2+}]$ and $[Co^{2+}]$, and the ability to use Co to restore Zn-limited growth was species-specific and consistent with prior results for the diatoms *T. pseudonana, P. tricornutum* and *P. delicatissima* (Fig. 1a, b)[9] and for other eukaryotic algae[2,8,10]. Growth rates of *Chaetoceros* RS19 were not stimulated by increasing $[Co^{2+}]$ up to 23.5 pM in the absence of added Zn. This inability to substitute Co for Zn in *Chaetoceros* RS19 was clearly distinct from that of other diatoms, but was consistent

with previous observations in *Chaetoceros calcitrans*[10], implying a genus-wide attribute.

The proteome as a function of $Zn^{2+}$ and $Co^{2+}$ was explored in the marine diatom *T. pseudonana* harvested during log phase growth. Global proteomic analysis comparing low (1.1 pM) versus high (10.2 pM) added $[Zn^{2+}]$ and low (2.3 pM) versus high (23.4 pM) added $[Co^{2+}]$ revealed two uncharacterized diatom proteins that greatly increased in abundance at low $[Zn^{2+}]$ or $[Co^{2+}]$ (Fig. 1c, d). These proteins were annotated as a CobW/HypB/UreG, nucleotide binding domain and a bacterial extra-cellular solute binding domain, respectively, within the manually curated JGI Thaps3 *T. pseudonana* genome[17] and were identified in *T. pseudonana* cultures with high confidence (≥9 exclusive unique peptides, 100% protein probability; Supplementary Fig. 1). BLAST sequence alignments showed these proteins to be homologous with CobW-like proteins (with 31.69% identity relative to *Pseudomonas denitrificans* CobW) and with the bacterial nickel transport protein NikA (with 30.5% identity relative to *E. coli* NikA), respectively. Based on their clear response to Zn and Co in the proteomes of multiple diatom species (Fig. 2a–d), the lack of definitive annotations in diatoms, and their genetic distance from bacterial homologs, these proteins are referred to as ZCRP-A and ZCRP-B (Zn/Co Responsive Protein A and B) in this study. Abundance patterns of these proteins were also investigated in *P. tricornutum, P. delicatissima* and *Chaetoceros* RS19. ZCRP-A spectral abundance counts were significantly (Kendall correlation, $p < 0.05$) inversely related to the total amount of Zn or Co added to the culture media (Fig. 2a–d; Supplementary Fig. 2). A similar pattern was found for ZCRP-B in the diatoms *T. pseudonana, P. tricornutum*, and *P. delicatissima*, though ZCRP-B spectral counts were significantly inversely related only to Zn concentration (Supplementary Fig. 2, $p = 0.034$). Unlike the continuous decrease in ZCRP abundances with increasing metal concentrations observed in *T. pseudonana* and *Chaetoceros* RS19, ZCRP abundances in *P. tricornutum* and *P. delicatissima* at the lowest metal treatments were somewhat smaller compared to their abundances in the next highest metal treatments, possibly owing to increased cellular stress and reduction in overall metabolism at the lowest added metal treatment in these diatoms. Homologs of ZCRP-B were not detected in the proteome of *Chaetoceros* RS19 (Fig. 2d). We note that the transcriptome of *Chaetoceros* RS19 used for peptide-to-spectrum mapping of the proteome also lacked ZCRP-B, which implies that either the conditions the transcriptome were generated under (iron limitation) did not elicit a low Zn response and corresponding ZCRP-B transcripts, or alternatively, that *Chaetoceros* RS19 does not possess the *ZCRP-B* gene. As a result, proteomic nondetection of ZCRP-B in *Chaetoceros* RS19 in this study cannot currently be interpreted due to its potential absence from the genome.

Among the four diatom species, both ZCRPs became undetectable at different threshold divalent metal cation concentrations. For *T. pseudonana*, both proteins became undetectable at 10.2 pM $Zn^{2+}$, but both were still detected in the presence of up to 23.4 pM $Co^{2+}$ (Fig. 2a). The threshold $Zn^{2+}$ concentration for *P. tricornutum* was lower, as neither protein was detected in 3.2 pM $Zn^{2+}$. A threshold $Co^{2+}$ concentration also uniquely existed in *P. tricornutum*, with neither protein detected in 23.4 pM $Co^{2+}$ (Fig. 2b). Similar to *T. pseudonana*, both proteins in *P. delicatissima* were detectable up to 23.4 pM $Co^{2+}$, and a Zn threshold for ZCRP-B was observed at 10.2 pM $Zn^{2+}$ (Fig. 2c). Overall, $Zn^{2+}$ thresholds eliciting detectable ZCRP abundances among these diatoms were lower compared to $Co^{2+}$ thresholds. ZCRP-B was not detected in *Chaetoceros* RS19, and ZCRP-A was still detectable up to 10.2 pM $Zn^{2+}$ (Fig. 2d). No proteomic data is available for *Chaetoceros* RS19 grown in Co amended media as this diatom is incapable of substituting $Co^{2+}$

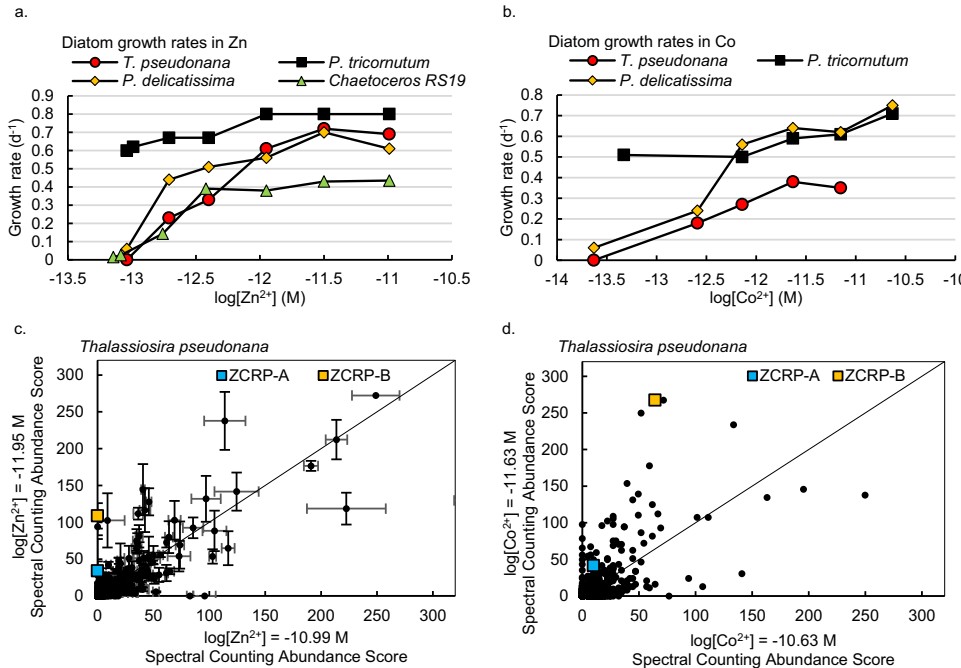

**Fig. 1 Growth responses of diatoms to varying [Zn$^{2+}$] and [Co$^{2+}$] and initial detection of ZCRPs in *T. pseudonana*.** Growth rates of four diatoms over a range of **a** [Zn$^{2+}$] and **b** [Co$^{2+}$]. Data are presented as mean values of biological duplicate cultures. Data is available in Supplementary Table 1. Global proteomic analyses comparing the proteomes of pooled biological duplicate cultures ($n = 2$) of *T. pseudonana* in **c** high vs. low added Zn and **d** high vs. low added Co. Each point is an identified protein with the mean of technical triplicate abundance scores in one treatment plotted against the mean of abundance scores in another treatment. The solid line denotes 1:1 abundance. Error bars in **c** are the standard deviation of technical triplicate measurements.

for Zn$^{2+}$, thus obtaining biomass under Co growth was not possible. An inverse relationship between spectral counts of both proteins and growth rate was apparent in *P. tricornutum* across +Zn and +Co treatments, with protein expression increasing as growth rate decreased (Fig. 2e, f; Supplementary Table 1).

**Putative metal binding capability of ZCRP-A and cellular localization**. Sequence alignment (BLASTp) of the ZCRP-A proteins detected in *T. pseudonana, P. tricornutum, P. delicatissima*, and *Chaetoceros* RS19 revealed the presence of four conserved motifs that classify these proteins into the COG0523 family of G3E GTPases. These include the G1/Walker A, G2/Switch I, putative metal-binding CXCC (C, cysteine; X, any amino acid, often hydrophobic), and G3/Walker B motifs[8,13] (Fig. 3a). Predicted tertiary structures of the ZCRP-A homologs in *T. pseudonana, P. tricornutum, P. delicatissima*, and *Chaetoceros* RS19 were modeled in SwissModel[18] using *E. coli* YjiA as a template (PDB entry 4IXM.2.b) to investigate putative Zn$^{2+}$ binding capacity. We included the *E. coli* COG0523 GTPase YjiA as it is the only member of the COG0523 subfamily to date with a determined crystal structure[13]. YjiA is currently thought to be a metal chaperone[8,19]. We furthermore included as a template a COG0523 protein previously described in the freshwater marine alga *Chlamydomonas reinhardtii* that was observed to have increased transcript expression under Zn-deficient conditions[8].

With the exception of the ZCRP-A homolog in *Chaetoceros* RS19, all ZCRP-A homologs were predicted to possess at least one Zn$^{2+}$ binding site formed by coordination to two glutamate (E) and one cysteine (C) residue (Fig. 3b), the cysteine residue belonging to the conserved putative metal-binding CXCC motif (Fig. 3a, b). Previous work with bacterial CobW has shown that high affinity metal binding requires the association of the protein with magnesium (Mg$^{2+}$) and GTP, with the thiol-containing Co$^{2+}$ binding site in Mg$^{2+}$GTP-CobW thought to be derived

from the CxCC motif and a tetrahedral geometry[19]. While no model of a Mg$^{2+}$GTP-bound COG0523 protein yet exists, the presence of GTP-binding domains in diatom ZCRP-A (Fig. 3a) and a Zn$^{2+}$ coordination environment comparable to that of YjiA suggests that high affinity metal binding in diatom ZCRP-A also likely requires association with Mg$^{2+}$ and GTP. Furthermore, while the model suggests that the Zn$^{2+}$ coordination environment of ZCRP-A is comparable to that of YjiA, this does not mean that ZCRP-A and YjiA necessarily fulfill the same cellular function since the C-terminal domain, which is thought to be an indicator of protein–protein interaction specificity, is quite different in these two proteins.

The general cellular localization of ZCRP-A was also explored using overexpression (OE) lines of yellow fluorescent protein (YFP) tagged ZCRP-A in *P. tricornutum* (ZCRPA-OE). ZCRP-A appears to be an intracellular protein that localizes in proximity to the endoplasmic reticulum (ER) (Fig. 4a and Supplementary Fig. 3a). The intracellular distribution of YFP alone was distinctly different from the distribution of YFP-tagged ZCRP-A and YFP-tagged ZCRP-B (Supplementary Fig. 4). Diatoms possess complex plastids with four distinct membranes, the outermost being contiguous with the ER[20]. Given that the *P. tricornutum* ZCRP-A sequence lacks a predicted plastid signal peptide (but rather has a predicted cytosolic signal peptide), we hypothesize that ZCRP-A localizes to the ER or the portion of the contiguous ER membrane directly adjacent to the plastids, the chloroplast ER (CER).

To date, connections between COG0523 proteins and utilization of Zn and Co have been explored primarily in prokaryotic organisms. For example, the COG0523 protein CobW has a role in vitamin B$_{12}$ biosynthesis and thus Co use[19,21]. In contrast, a subgroup of other COG0523 proteins (YjiA, YeiR, ZigA, and ZagA) have been implicated in Zn$^{2+}$ metabolism[8,13–16], and a client protein to the metallochaperone ZagA in *Bacillus subtilis* has been identified[22].

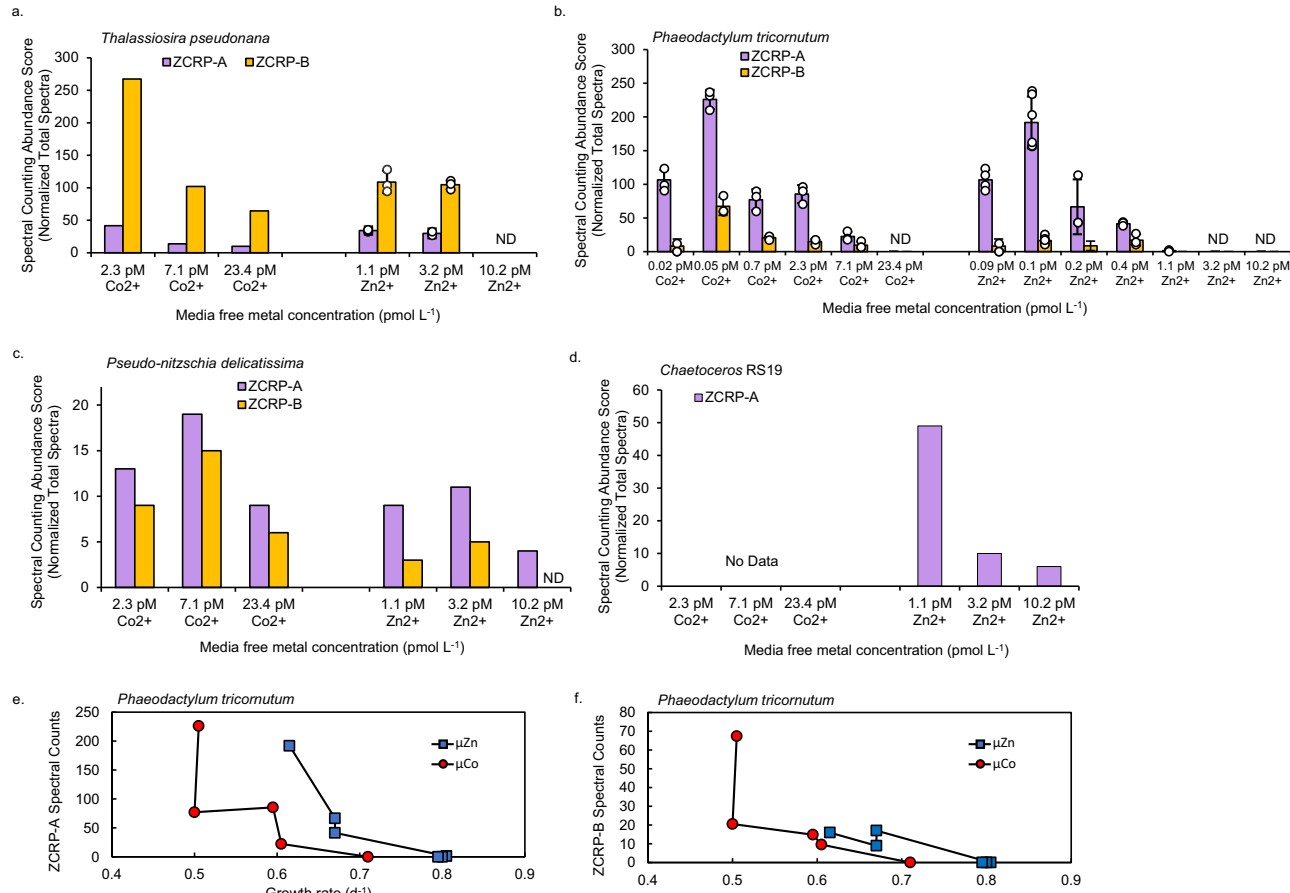

**Fig. 2 Expression trends of ZCRPs in four marine diatom species.** Mean spectral counting abundance scores of ZCRP-A and ZCRP-B in **a** 6 treatments of *T. pseudonana*, **b** 12 treatments of wild-type *P. tricornutum*, **c** 6 treatments of *P. delicatissima,* and **d** 3 Zn treatments of *Chaetoceros* RS19 as measured by global proteomic analysis. For all diatoms, spectral counts were derived from pooled biological duplicate cultures ($n = 2$). Where available, data are presented as mean values ± the standard deviation of technical triplicate measurements with individual data points overlaid (white circles). Data presented without error bars were measured in technical singlicate. ND, not detected. No proteomic data is available for *Chaetoceros* RS19 grown in Co treatments as ample biomass for proteomic analysis was not available. **e** Spectral counts of ZCRP-A and **f** ZCRP-B compared to growth rates of *P. tricornutum* in Co treatments (μCo) and Zn treatments (μZn). Data available from Supplementary Table 1.

Compared to bacteria, less is known about the function of COG0523 proteins in marine phytoplankton, though COG0523 protein family members are known to occur in all kingdoms[8,23]. A recent study described the presence of COG0523 domain proteins upregulated under low Zn in the coccolithophore *Emiliania huxleyi*, but without further functional characterization[24], implying a potential Zn-related function of a COG0523 protein in a marine alga distinct from the marine diatoms included in this study.

Although various proteins belonging to the COG0523 subgroup share similar conserved domains, they possess different metal binding abilities and thus likely have different functions among the diverse organisms in which they are found. For example, recent work has established that CobW preferentially binds $Co^{2+}$ as the cognate metal and acts as a $Co^{2+}$ chaperone ultimately supplying vitamin $B_{12}$ in bacteria, whereas the closely related putative metal chaperones YeiR and YjiA (homologs of CobW) bind $Zn^{2+}$[19]. We can infer from homology and the response to low Zn and low Co in the present study that $Zn^{2+}$ and $Co^{2+}$ are likely both cognate metals for diatom ZCRP-A. Further metal binding and affinity assays can confirm and characterize metal binding in this protein.

**Frustule morphology**. Phenotypic plasticity in *P. tricornutum* is well documented. Two basic cell morphotypes, fusiform and triradiate, are found in natural liquid environments. It is thought that by adopting the triradiate form, a cell increases its surface area and thus the area of membrane available for enzymatic activity or molecular diffusion of dissolved inorganic carbon (DIC) into the cell. The triradiate form is known to be more common under DIC limiting conditions, which supports this hypothesis[25]. Distinct morphological differences resulted from the knockout (KO) of the ZCRP-A gene. In *P. tricornutum*, ZCRP-A knockout cells consistently adopted a triradiate shape while wild-type cells were fusiform (Fig. 4i). Normally, triradiate cells of *P. tricornutum* spontaneously revert to fusiform across generations[26], thus it is notable that ZCRP-A knockout cells have consistently maintained their triradiate shape for 4+ years in culture irrespective of media [$Zn^{2+}$]. As $Zn^{2+}$ is the predominant metal cofactor used in diatom CAs, the adoption of the triradiate form in knockout *P. tricornutum* cells may be a response to a disruption of the carbon concentrating mechanism caused by a reduction in Zn acquisition capability due to ZCRP-A knockout. This is consistent with the observed relative increase in $Mn^{2+}$-utilizing CA (ι-CA) in the knockout line compared to the wild-type (Supplementary Fig. 5).

**ZCRP-B sequence analysis and cellular localization**. Unlike COG0523 proteins, the relationship of ZCRP-B abundance to

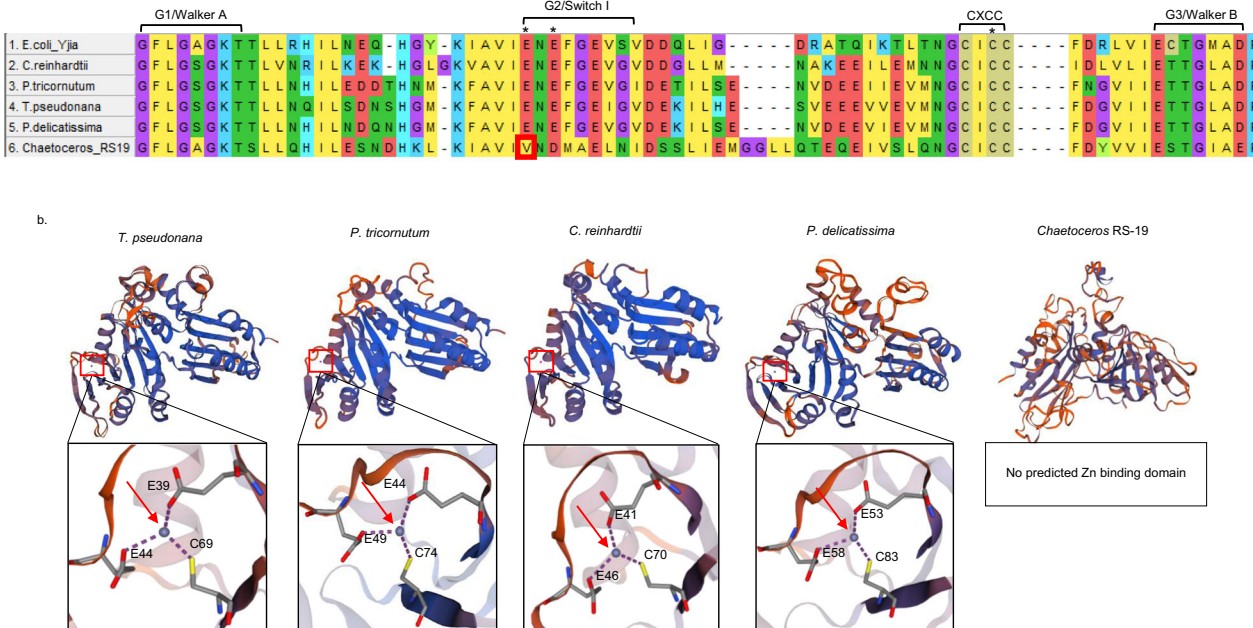

**Fig. 3 Algal ZCRP-A sequence alignments and predicted structures. a** Sequence alignment of the *E. coli* YjiA protein compared to ZCRP-A proteins in *C. reinhardtii*, *P. tricornutum*, *T. pseudonana*, *P. delicatissima*, and *Chaetoceros* RS19 generated using MUSCLE. Four conserved GTPase domains including the putative CXCC metal binding motif are labeled. Asterisks denote the glutamate (E) and cysteine (C) residues predicted to bind $Zn^{2+}$. Notably, one of the metal binding E resides in the G2/Switch I region has been replaced by valine (V; red box) in *Chaetoceros* RS19. **b** Predicted structures of ZCRP-A homologs modeled using the well-characterized $Zn^{2+}$ and $Co^{2+}$ binding COG0523 GTPase YjiA protein in *E. coli* as a template (PDB entry 4IXM.2.b). The predicted ligand $Zn^{2+}$ is indicated by a red arrow. Close-up views show the predicted $Zn^{2+}$ binding site composed of two glutamate (E) and one cysteine (C) residue.

environmental Zn and Co concentrations does not appear to have been previously described. Topology predictions of *P. tricornutum* ZCRP-B using TOPCONS[27,28] revealed a single predicted transmembrane domain near the N-terminus, with the majority of the protein predicted to be oriented outside the membrane (Fig. 4j). Overexpression and fluorescent tagging of ZCRP-B confirmed localization to the cell membrane (Fig. 4e–h; Supplementary Fig. 3b). A single predicted transmembrane domain contrasts with the Zrt/Irt-like divalent metal transporters (ZIPs) in eukaryotic algae, which have 7+ transmembrane domains and are key Zn transporters in many organisms[29,30]. It is therefore most likely that ZCRP-B is not a transporter itself, but one part of a multi-protein membrane complex and potentially interacts with the ZIP system. A sequence database similarity search (BLASTp, NCBI) found the ZCRP-B protein to be homologous with NikA, a protein subunit of the bacterial ATP-binding cassette (ABC) type Ni transport system protein Nik (30.5% identity with *E. coli* NikA, $E = 7e-49$, Supplementary Fig. 6). This transporter is well characterized in bacteria and is comprised of five subunits NikA-E. NikB and NikC are two pore-forming integral inner membrane proteins, NikD and NikE are two inner membrane-associated proteins with ATPase activity, and NikA is the periplasmic component that functions as the initial metal receptor[31]. No proteins with homology to NikB nor NikC were detected in the *P. tricornutum* proteomes generated in this study. Two uncharacterized *P. tricornutum* proteins were homologous with NikD (28.8% identity, $E = 1e-14$) and NikE (34.9% identity, $E = 1.33e-8$), though neither had abundance trends similar to ZCRP-B, implying that their function and regulation are independent of ZCRP-B.

The sequence of a functionally similar bacterial ABC transport complex, CntABCDF (cobalt nickel transporter, also known as Opp1) from *Staphylococcus aureus* was also compared to NikA and ZCRP-B (Supplementary Fig. 6). CntA shares 25.6% identity with ZCRP-B ($E = 3e-28$), and similar to NikA, is an extra-cytoplasmic solute-binding protein that transports Ni, Zn and Co.

CntA functions as a Ni/Co acquisition system in Zn-limited *S. aureus*[32]. Although the Nik and Cnt systems serve Ni and Co transport in bacteria, ZCRP-B responds to Zn and Co in marine diatoms, which have a significant Zn demand. This may imply a recruitment and repurposing of this bacterial Ni transporter component as part of the Zn acquisition systems during the evolution of marine diatoms.

**ZCRP-B as a putative high-affinity ligand.** Sequence similarity to the extracellular transport components NikA and CntA (Supplementary Fig. 6), localization to the plasma membrane (Fig. 4b; Supplementary Fig. 3b), and increased abundance under low Zn and Co conditions (Fig. 2b) of *P. tricornutum* ZCRP-B suggests a metal-binding role as part of a high-affinity transport complex. The induction of ZCRP-B expression at low $[Zn^{2+}]$ (Fig. 2a–c) fits the description of a high-affinity Zn uptake system observed in marine algae that is known to be induced at low free $[Zn^{2+}]$[33,34], suggesting that this protein is involved in an adaptive response to extremely scarce Zn availability. Furthermore, ZCRP-B could contribute to the pool of high-affinity organic ligands that complex dissolved Zn, either by dissociation from living cells or upon cell death by viral lysis and grazing, in the upper water column[12,35].

The identification of a membrane-associated Zn-Co responsive protein-containing putative metal-binding sites allows us to reconsider the mechanisms of cellular metal uptake in diatoms. Prior physiological experiments observed Zn uptake in marine diatoms to approach the limits of diffusion[33], and predicted kinetic control with fast cell surface metal binding and uptake relative to dissociation and release back to the seawater environment[36]. To enable this transport capability, it was postulated that transporters might be so abundant that the membrane becomes crowded[37]. Here, the observation of a putative Zn-binding, membrane-associated protein with only 1

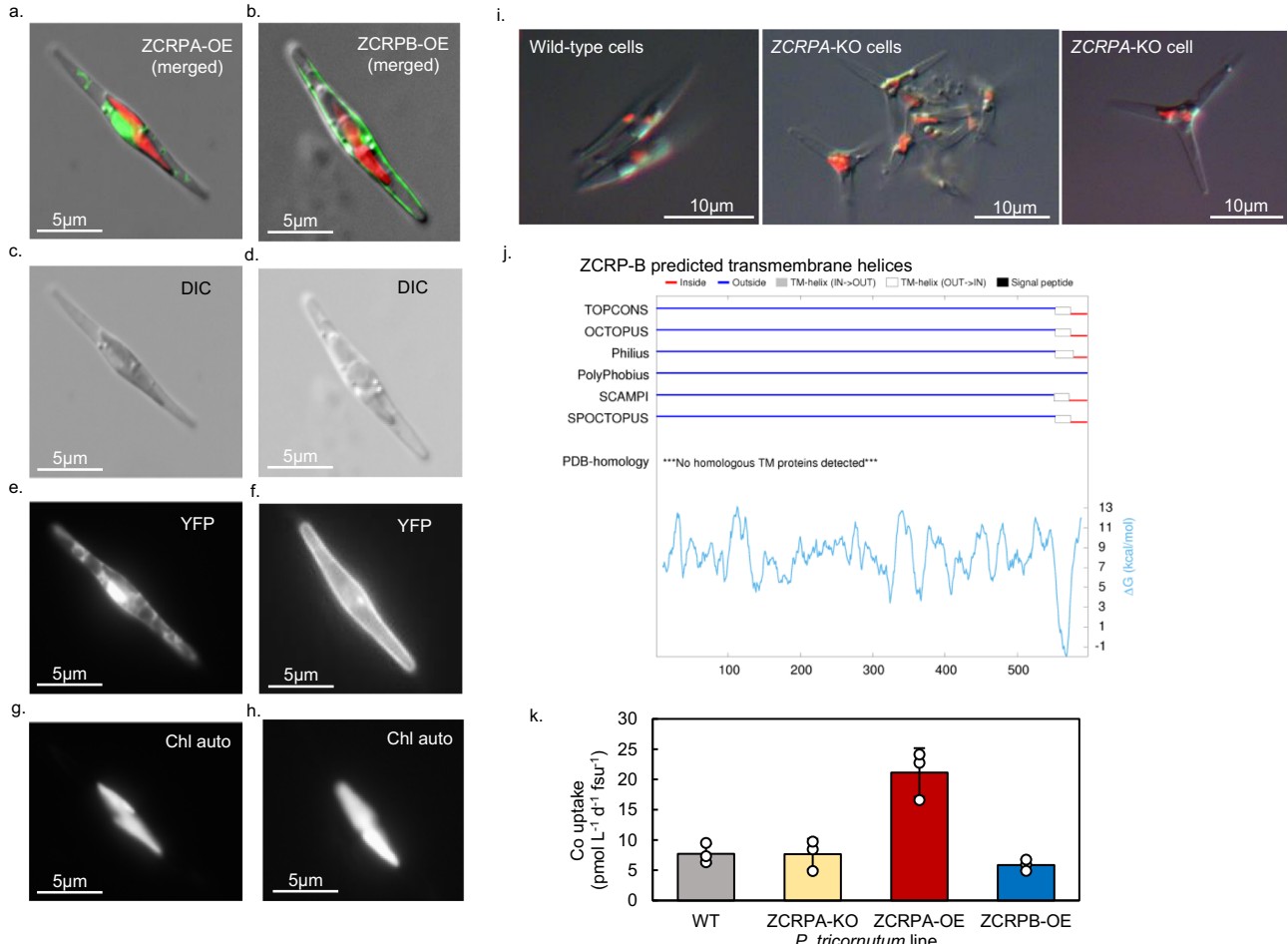

**Fig. 4 Confocal and epifluorescent micrographs of the ZCRP-A and ZCRP-B proteins fused to YFP and overexpressed in *P. tricornutum*. a** Localization of ZCRP-A to the chloroplasts and **b** localization of ZCRP-B to the cell membrane. YFP fluorescence is false-colored green and chlorophyll autofluorescence is false-colored red. Composite images are stacks of the individual channels **c**, **d** differential interference contrast (DIC), **e**, **f** yellow fluorescent protein (YFP), and **g**, **h** chlorophyll autofluorescence (Chl auto). **i** Micrographs of Zn-limited wild-type and *ZCRP-A* knockout (KO) *P. tricornutum* cells showing morphological differences. For **a**–**h** and **i**, results were validated > 10 times. **j** Topology predictions from five sub-methods (OCTOPUS, Philius, PolyPhobius, SCAMPI, and SPOCTOPUS), consensus prediction (TOPCONS), and predicted $\Delta G$ values for *P. tricornutum* ZCRP-B generated using the TOPCONS webserver (https://topcons.cbr.su.se/)[27,28]. **k** Extent of Co uptake after 24 h for wild-type (WT), ZCRPA-knockout (KO), and ZCRPA-overexpression (OE) lines of *P. tricornutum* normalized to fluorescence units (fsu). Data are presented as mean values ± the standard deviation of biological triplicate cultures (*n* = 3). Individual data points are overlaid as white circles. The extent of Co uptake was found to be significantly larger in the ZCRPA-OE line compared to the wild-type via one-way ANOVA (f(3) = 23.16, *p* = 0.000268) and post hoc Dunnett test (*p* = 0.00048).

predicted transmembrane domain instead implies a separation of the Zn concentrating function at the cell surface relative to its transport into the cell. In this scenario when Zn is scarce, biosynthesis of ZCRP-B increases and is tethered to the cell surface to compete Zn away from natural dissolved Zn ligands[35] and/or chelate Zn atoms that make it through the diffusive boundary layer to the membrane. In this manner, ZCRP-B would increase the surface Zn concentration in the vicinity of Zn transporters, and multiple ZCRP-B proteins could supply nearby surface ZIP transporters or be endocytosed, avoiding the predicted membrane crowding of transporters problem. Aristilde and colleagues have previously demonstrated that weak natural Zn-binding ligands containing cysteine do indeed enhance cellular Zn uptake within the diatom *Thalassiosira weissflogii*, with heightened effects in Zn-limited compared to Zn-replete cells[38]. They proposed the formation of a transient tertiary complex between the Zn-bound ligand and Zn transporters (ZIPs and heavy metal P-type ATPases) at the cell surface, which could be mediated by a surface-tethered Zn binding ligand such as ZCRP-B. Future studies could examine the mechanism of Zn exchange between ZCRP-B and Zn/Co transporters such as the ZIPs in eukaryotic algae, which were also detected at lower Zn and Co abundances in *P. tricornutum* but with relatively lower spectral counts (Supplementary Fig. 7a, b), consistent with this model. Furthermore, the proposed mechanism of ZCRP-B binding is similar to that of the high-affinity $Fe^{3+}$ binding protein ISIP2a, previously characterized in marine algae as an iron starvation-induced protein[39]. ISIP2a has been characterized as a phytotransferrin involved in endocytosis-mediated high-affinity Fe uptake in *P. tricornutum* that acts to concentrate Fe at the cell surface and is an extracellular protein anchored to the membrane with one transmembrane domain[39]. As the protein sequences of *P. tricornutum* ZCRP-B and ISIP2a share no significant similarity, it is possible that the uptake mechanism of ZCRP-B is similar to that of ISIP2a, but specific to high-affinity Zn and Co uptake rather than Fe. This suggests a common strategy of using extracellular membrane-anchored metal acquisition proteins in marine algae faced with metal limitation.

**Co uptake in wild-type and mutant diatom strains**. As ZCRP-A and ZCRP-B abundance is related to media $[Co^{2+}]$ (Fig. 2a–d), we investigated differences in the extent of Co uptake after 24 h among Zn/Co-limited wild-type, ZCRP-A knockout, ZCRP-A overexpression, and ZCRP-B overexpression lines of *P. tricornutum* via addition of the radiotracer $^{57}Co$ (see methods). The extent of Co uptake among genetically modified *P. tricornutum* lines was observed to be significantly different via one-way ANOVA $(f(3) = 23.16, p = 0.000268)$. A Dunnet post hoc test revealed that uptake was significantly greater (2.6× larger) in the ZCRP-A overexpression line compared to wild-type $(p = 0.00048,$ Fig. 4k). We interpret this result as the overexpression of ZCRP-A creating a larger intracellular binding capacity for Co, thus protecting it from intracellular sensor or regulatory systems and/or efflux pumps. In contrast, no significant difference in Co uptake rates was observed when comparing ZCRP-A knockout, ZCRP-B overexpression, and wild-type lines, suggesting that *P. tricornutum* ZCRP-A knockout cells are capable of compensating for knockout to maintain Co metabolism, perhaps through the use of low-affinity transporters[33]. This is consistent with these uptake experiments being conducted using seawater media with a relatively abundant concentration of Zn (background of 7.7 pM Co and 4.0 nM Zn in the absence of EDTA), thus the use of low-affinity transporters was likely sufficient to acquire Zn and Co for growth, and neither ZCRP-A knockout nor ZCRP-B overexpression would be expected to add any metabolic benefit (Fig. 4k). Moreover, if ZCRP-B is only one part of a multi-protein acquisition and transport complex as hypothesized, overexpression of the single protein may not result in enhanced functionality.

**Abundance patterns of CAs in two diatoms**. Carbonic anhydrase enzymes constitute a major reservoir of Zn and Co within marine diatoms[7]. Within the stroma, intracellular chloroplastic CAs are essential in supplying $CO_2$ to RUBISCO as they convert $HCO_3^-$, the predominant species of inorganic carbon in the pyrenoid, into $CO_2$[40,41]. Seven subclasses of CAs have been identified in marine diatoms to date and are designated as alpha, beta, gamma, delta, zeta, theta, and iota (α, β, γ, δ, ζ, θ, and ι). While $Zn^{2+}$ is the cofactor most commonly used in algal CAs, utilization of both cadmium ($Cd^{2+}$) and cobalt ($Co^{2+}$) in place of $Zn^{2+}$ at the active site of ζ-CA (CDCA) and a δ-CA, respectively, has been previously documented[2,5,42]. Overall, Zn-utilizing CAs increased in abundance with increasing Zn, consistent with the need for rapid $HCO_3^-$ conversion at faster growth rates (Fig. 5; Supplementary Fig. 7). Specifically, spectral abundance counts of two β-type CAs, PtCA1 and PtCA2, became abundant in high $[Co^{2+}]$ (23.4 pM) and $[Zn^{2+}]$ (> 1.1 pM) and were inversely related to ZCRP-A abundance (Supplementary Fig. 7). Both PtCA1 and PtCA2 are known to localize to the chloroplast pyrenoid[41,43]. Moreover, the increasing abundance trends of the Zn-utilizing α-CAs (CA-II and CA-VI) and the θ-CA Pt43233, which localize to the periplastidial compartment, chloroplast endoplasmic reticulum, and thylakoid lumen, respectively, at higher and Zn/Co provide further evidence for this strategy of increasing CA use under Zn-replete and higher growth rate conditions (Fig. 5; Supplementary Fig. 7)[43,44].

In contrast, abundance trends of the recently discovered ι-CA were inversely related to $Zn^{2+}$ (Fig. 5). Originally identified in *T. pseudonana*, ι-CA was found to localize to the inner chloroplast membrane surrounding the stroma and is unusual in that it prefers $Mn^{2+}$ to $Zn^{2+}$ as a cofactor[45]. In the present study, spectral counts of *P. tricornutum* ι-CA decreased as metal concentrations increased, similar to that observed for ZCRP-A and ZCRP-B (Fig. 5). This ι-CA response was consistent with a Zn sparing strategy under low $[Zn^{2+}]$ and $[Co^{2+}]$ used to prioritize the use of $Zn^{2+}$ for other metalloenzyme functions.

Due to the inverse relationship between the abundances of ZCRP-A and chloroplastic $Zn^{2+}$-requiring CAs in *P. tricornutum* (that is, all CAs detected with the exception of ι-CA) and the various types of CAs in *T. pseudonana* (Supplementary Fig. 7), it seems unlikely ZCRP-A directly interacts with CAs. These results are instead consistent with the hypothesis that ZCRP-A functions as a $Zn^{2+}$ allocation and prioritization mechanism during Zn limitation. The role of $Zn^{2+}$ in key transcriptional and translational proteins such as RNA polymerase and ribosomal proteins is well known, and major reservoirs of Zn are associated with these transcription and translation systems in the fast-growing copiotrophic bacterium *Pseudoalteromonas*[6]. The availability of Zn in ribosomes and the ER is therefore likely also a cellular priority in diatoms, and could benefit from utilizing the putative chaperone and trafficking capability of ZCRP-A when Zn is scarce. We, therefore, posit that ZCRP-A may serve as a $Zn^{2+}$ trafficking or storage protein that contributes to the prioritization and movement of $Zn^{2+}$ to the ER or CER, while the Mn-utilizing Mn ι-CA compensates for the lowered Zn availability in the chloroplast. The increased biosynthesis of ZCRP-A may be an important function to shift Zn homeostasis, competing for intracellular Zn and trafficking it towards the ER or CER.

**Distribution of putative ZCRP homologs among oceanic taxa**. Putative ZCRP homologs among eukaryotic oceanic taxa were identified by BLAST searching the *P. tricornutum* ZCRP-A and ZCRP-B protein sequences against all available transcriptomes in the Marine Microbial Eukaryotic Transcriptome Sequencing Project (MMETSP) database, which includes over 650 assembled and annotated transcriptomes of oceanic microbial eukaryotes[46]. Phylogenetic analysis revealed the presence of putative ZCRP-A and ZCRP-B homologs in a wide variety of organisms belonging to the Chromista kingdom that could be further categorized into Bacillariophyceae, Dinophyceae, and Prymnesiophyceae classes (Supplementary Figs. 8 and 9). Notably, the *Chaetoceros* RS-19 ZCRP-A homolog did not phylogenetically cluster with the other diatoms (Bacillariophyceae), but instead appears to be more closely related to *E. coli* YjiA (Supplementary Fig. 8). Furthermore, the lack of the conserved G2/Switch I region in the *Chaetoceros* RS-19 homolog (Fig. 3) is anomalous in comparison to other putative homologs identified within the MMETSP database. Overall, ZCRPs are not exclusive to oceanic diatoms, but rather are widely distributed amongst oceanic taxa.

**Metaproteomic detection of ZCRP-A and ZCRP-B**. To investigate the use of ZCRP-A and ZCRP-B in the natural environment, we searched metaproteomic data collected during the KM1128 METZYME (Metals and Enzymes in the Pacific) research expedition on the R/V *Kilo Moana* October 1–25, 2011 from Oʻahu, Hawaiʻi, to Apia, Samoa (Fig. 6a). dZn followed a nutrient-like distribution as described previously, with an average surface (40 m) dZn concentration of 1.21 nM and average deep water (3000 m) concentration of 10.37 nM[47] (Fig. 6b). dCo was highly depleted in the upper photic zone as the result of biological uptake[48,49] (Fig. 6c). Eukaryotic homologs of ZCRP-A and ZCRP-B were detected at multiple stations at surface (<200 m) depths with increased abundances moving southward. This was coincident with a deepening of the upper water column layer in which dZn and dCo were depleted (a "zincocline"), with depleted surface concentrations of dZn (and dCo) observed at greater depths south of the equator (Fig. 6b, c). The phytoplankton pigments chlorophyll *a*, fucoxanthin, and 19′-hexanoyloxyfucoxanthin (19′-Hex) are used as chemotaxonomic proxies for diatom

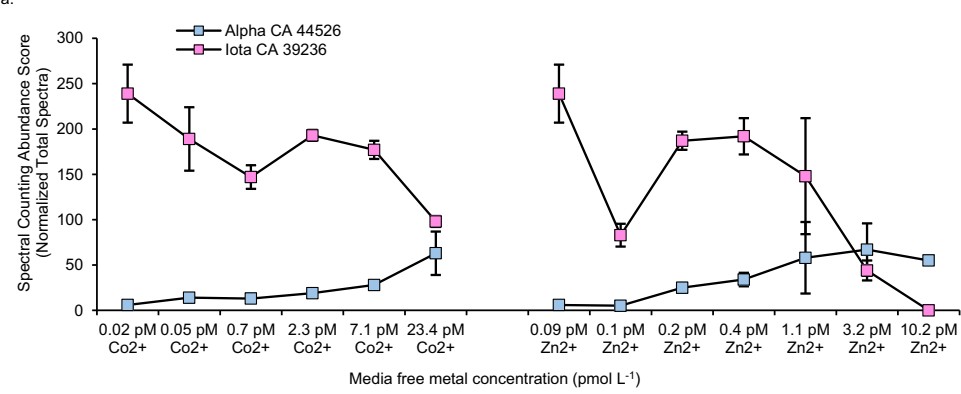

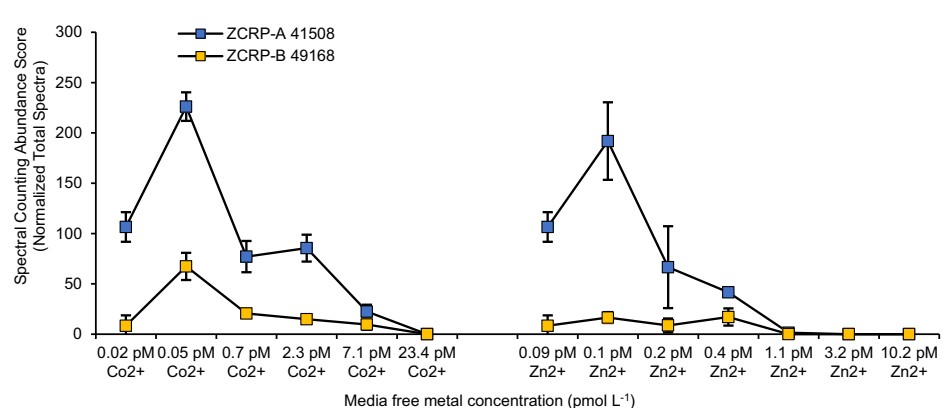

**Fig. 5 Comparison of α-CA, ι-CA, and ZCRP abundances.** Spectral counting abundance scores of **a** alpha CA, iota CA, and **b** ZCRP-A and ZCRP-B detected in Zn and Co treatments of *P. tricornutum* measured by global proteomic analysis. Data are plotted as means ± the standard deviation of technical triplicate measurements of pooled biological duplicate cultures ($n = 2$). Protein names are shown with their corresponding JGI protein ID.

and prymnesiophyte abundances[50]. Distinct differences in the distribution of pigments compared to that of ZCRP proteins over this transect demonstrated that increased ZCRP protein abundances cannot be simply ascribed to an increase in overall biomass (Figs. 6; 7).

Putative ZCRP-A homologs corresponded to nearest taxonomic matches to the diatom *Helicotheca tamensis*, the dinoflagellate *Azadinium spinosum*, and the prymnesiophytes *Emiliania huxleyi* and *Phaeocystis sp.* (Fig. 6d, f; Supplementary Table 2). Putative ZCRP-B homologs were identified in various eukaryotes corresponding to nearest taxonomic matches for the diatoms *Pseudo-nitzschia fraudulenta* and *Leptocylindrus danicus* and to the dinoflagellates *Azadinium spinosum*, *Gonyaulax spinifera*, and *Symbiodinium sp.* (Fig. 6e, g; Supplementary Table 2). Increased expression of these homologs occurred at low [dZn] (0–1.5 nM) at surface depths in the Equatorial and South Pacific (Fig. 6f, g; Supplementary Fig. 10), a region where Fe limitation was also documented[47,48]. The preponderance of eukaryotic ZCRP-A and ZCRP-B coupled to observations of Zn scarcity in the South Pacific implies the existence of marine eukaryotic protists responding to Zn stress in this area.

**Deployment of ZCRP-A and ZCRP-B as bioindicators of oceanic Zn stress.** While nitrogen, phosphate, and iron are widely recognized to be nutritionally important to phytoplankton[51], other elements such as Zn are rarely examined, likely due to methodological difficulties arising from pervasive Zn contamination, yet Zn is present in extreme scarcity in large tracts of the surface ocean[52]. Given the demonstrated use of Zn by eukaryotic phytoplankton, it has been hypothesized that Zn may

limit oceanic production rates and influence the global carbon cycle[5]. Laboratory-based studies with phytoplankton isolates maintained in culture have shown that $Zn^{2+}$ may act as a limiting nutrient for some species at concentrations of ~1 pM[2,4,33], and we have shown in the present study that expression of ZCRP-A and ZCRP-B occurs at $Zn^{2+}$ and $Co^{2+}$ concentrations ~1 pM (Fig. 2a–d). We explored the utility of field detection of these proteins and suggest that ZCRP-A and ZCRP-B may be useful candidates as biomarkers of Zn stress in the field, just as the detection of other protein biomarkers has been successfully applied to diagnose regions of nutrient stress[48]. Our observations of homologous proteins among multiple eukaryotic protists throughout the South Pacific imply that Zn scarcity may be far more prevalent than previously recognized, and that the role of Zn in influencing marine primary productivity may therefore be underestimated.

We have identified and characterized two Zn and Co responsive proteins, ZCRP-A and ZCRP-B, in the marine diatoms *T. pseudonana*, *P. tricornutum*, *P. delicatissima*, and *Chaetoceros* RS-19. These findings describe previously uncharacterized proteins that are involved in the metabolic response to Zn/Co metal limitation in marine diatoms, and are widespread in marine protists. A putative $Zn^{2+}$ metallochaperone role for ZCRP-A in the allocation and reallocation of Zn under low Zn conditions and characterization of ZCRP-B as a membrane-associated putative metal-binding protein belonging to a multi-protein transport complex best fit our observations. Increased abundance of ZCRP-B at low [$Zn^{2+}$] and [$Co^{2+}$] with nondetection at higher metal concentrations fits the description of the high-affinity uptake mechanism in marine algae previously observed in physiology experiments[33]. We have also shown that metaproteomic detection

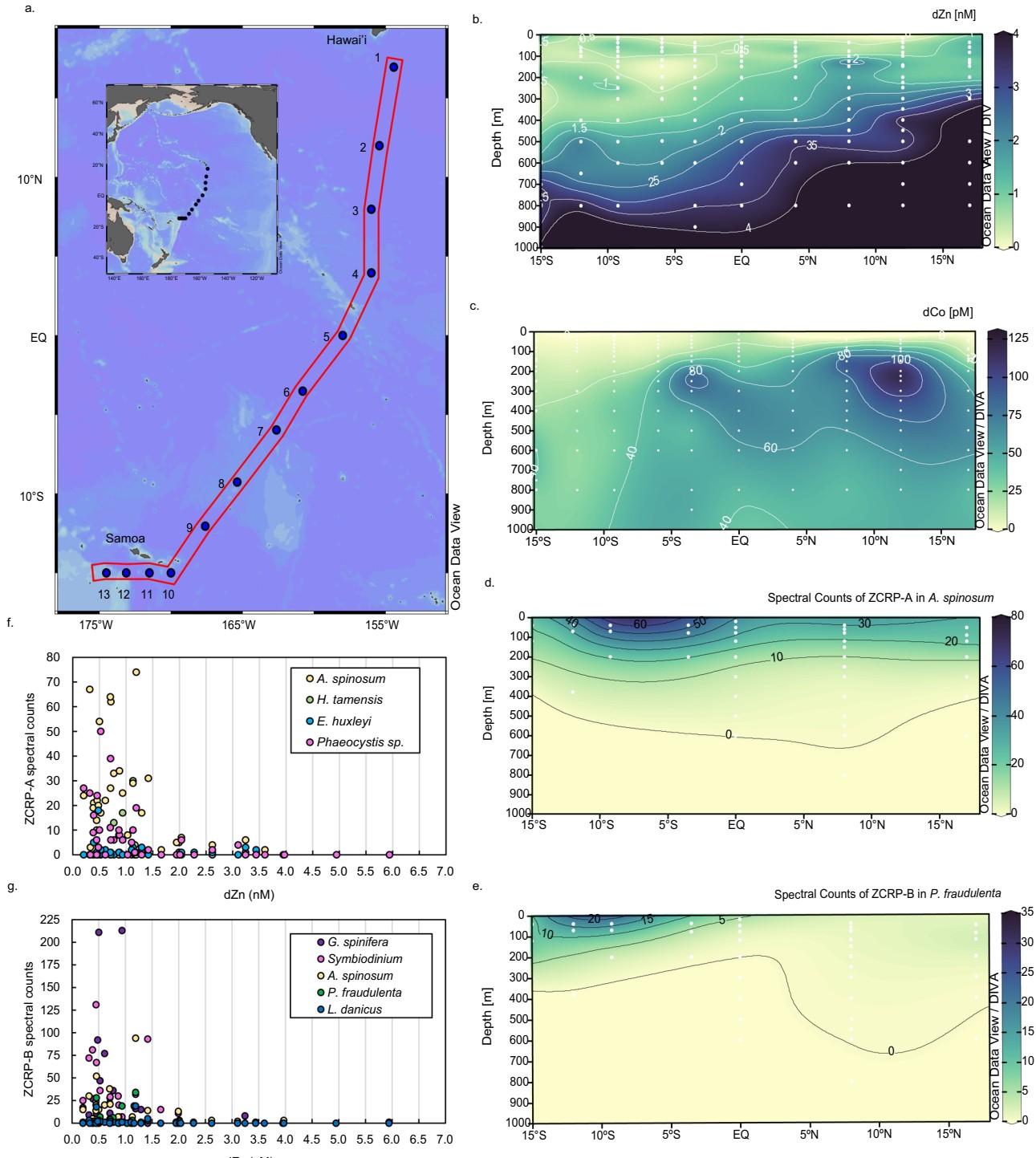

**Fig. 6 Metaproteomic detection of ZCRP homologs in the Southern Pacific. a** Transect sampled for dissolved Zn and metaproteomics during the KM1128 METZYME research expedition on the R/V *Kilo Moana* October 1–25, 2011 from Oʻahu, Hawaiʻi to Apia, Samoa, in the tropical South Pacific. **b** The concentration of total dissolved Zn and **c** the concentration of total dissolved Co measured in the upper 1000 m in color scale along the METZYME transect. **d** Total spectral counts of the ZCRP-A homolog in the dinoflagellate *Azadinium spinosum* and **e** total spectral counts of the ZCRP-B homolog in the diatom *Pseudo-nitzschia fraudulenta* and measured in the upper 1000 m in color scale along the METZYME transect. **f** Total spectral counts of ZCRP-A homologs detected in the dinoflagellate *A. spinosum*, the diatom *H. tamensis*, and the haptophytes *E. huxleyi*, and *Phaeocystis sp.* compared to total dissolved Zn concentration. **g** Total spectral counts of ZCRP-B homologs detected in the dinoflagellates *G. spinifera*, *Symbiodinium sp.*, *A. spinosum*, and in the diatoms *P. fraudulenta* and *L. danicus* compared to dZn. Detected species correspond to nearest taxonomic matches.

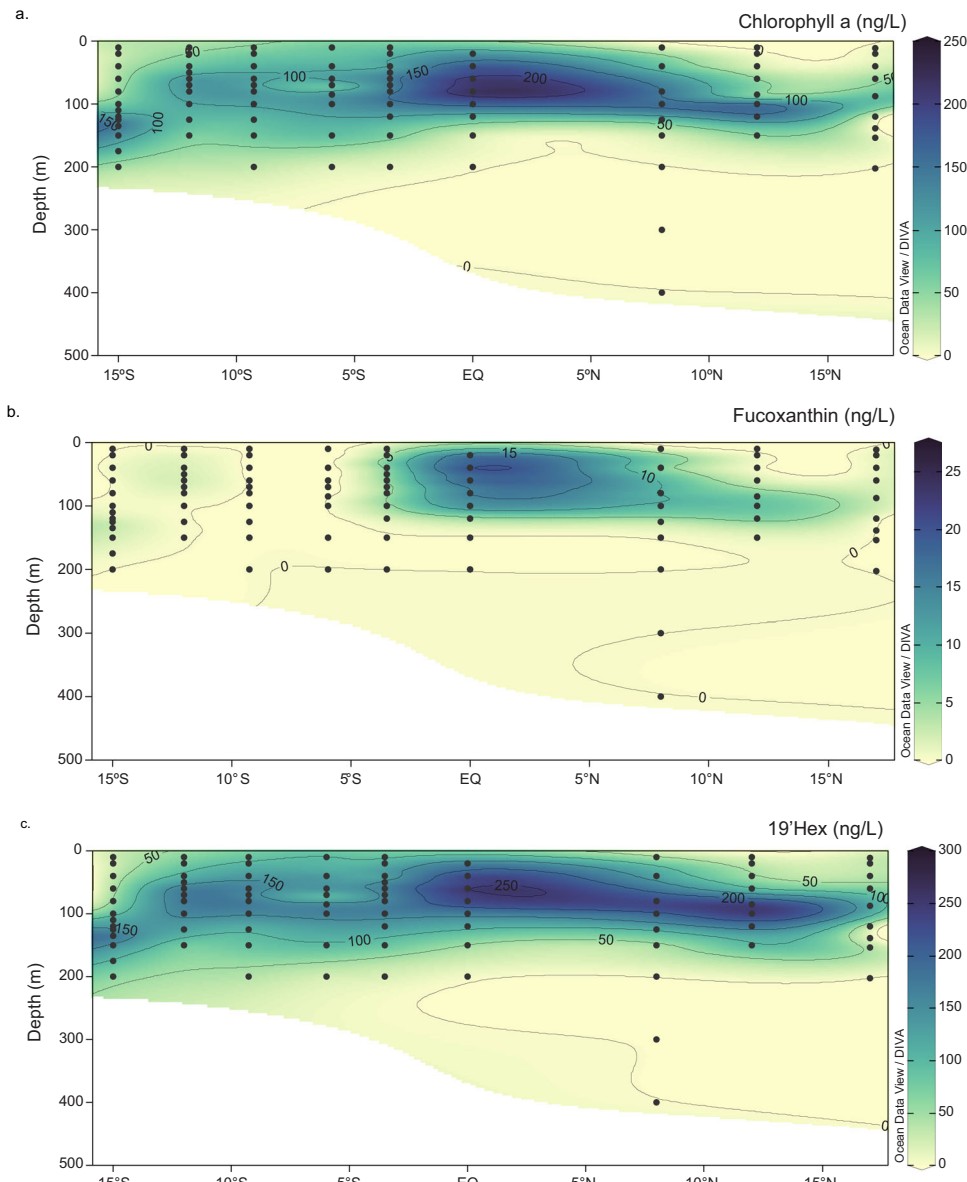

**Fig. 7 Phytoplankton pigment concentrations measured over the METZYME (KM1128) transect.** The concentration of **a** chlorophyll *a*, **b** fucoxanthin, and **c** 19'-hexanoyloxyfucoxanthin (19'-Hex) measured in the upper 500 m in color scale along the transect. This data was originally described in Saito et al. 2014 and Cohen et al. 2021 and is available on BCO-DMO (https://www.bco-dmo.org/project/2236).

of both ZCRP-A and ZCRP-B in the field is possible and that protein abundances are highest at the surface where dZn and dCo are scarcest, corroborating our findings in culture and demonstrating their potential for use in detecting Zn stress. Based on our characterization of these two Zn and Co responsive proteins in multiple marine diatoms in culture and the detection of homologs in the field, we suggest that Zn nutritional stress in the surface ocean could be more prevalent than previously thought.

## Methods

**Diatom species and culturing techniques.** *Thalassiosira pseudonana* CCMP1335 cultures were maintained in a 24 °C incubator under constant fluorescent lighting (65 μmol photon m$^{-2}$ s$^{-1}$). Cultures of both *Phaeodactylum tricornutum* CCMP632 and *Pseudo-nitzschia delicatissima* UNC1205 were maintained in a 18 °C incubator under constant fluorescent lighting (90 μmol photon m$^{-2}$ s$^{-1}$). *Chaetoceros sp.* RS-19 cultures were maintained at 4 °C under constant fluorescent lighting (40 μmol photon m$^{-2}$ s$^{-1}$). All lighting was provided by daylight white fluorescent bulbs. All cultures were randomly repositioned each day to avoid growth effects due to subtle variation in light intensity. *T. pseudonana* CCMP1335

and *P. tricornutum* CCMP632 (Bigelow Laboratory, East Boothbay, ME) were obtained from the Mincer and Saito laboratory culture collections at the Woods Hole Oceanographic Institution, respectively. *Chaetoceros* RS19 was originally collected from the Ross Sea, Antarctica, and isolated by D. Moran. Continuous culture maintenance was done in the Saito laboratory culture collection at the Woods Hole Oceanographic Institution. *P. delicatissima* UNC1205 was obtained from the Marchetti laboratory at the University of North Carolina. All cultures were axenic and maintained by sterile technique until needed.

Polycarbonate and plastic bottles were cleaned to remove trace metal contaminants before use. This procedure involved, at minimum, a 72 h soak in <1% Citranox detergent, five rinses in Milli-Q water, a 7-day soak in 10% HCl, and five rinses with dilute acid (HCl, pH 2). Cultures were grown in microwave sterilized 28 mL polycarbonate centrifuge tubes, and all solutions were pipetted only after a tip rinse procedure consisting of three rinses with 10% HCl followed by three rinses with sterile dilute HCl (pH 2). All culture work was conducted in a Class 100 clean room.

Culture media was prepared after Sunda and Huntsman for trace metal experimentation[2]. Microwave sterilized, 0.2 μm-filtered Pacific seawater from the North Pacific station Aloha (22° 45′N, 158° 00′W) was used as the media base. Macronutrients were added to this sterile base to a final concentration of 88.2 μM NaNO$_3$, 41.5 μM NaH$_2$PO$_4$, and 106 μM Na$_2$SiO$_3$ and were chelexed before use. Added vitamins included 2 nM biotin, 0.37 nM B$_{12}$ as cyanocobalamin, and 300 nM

thiamine and were also chelexed before use. Trace metals were added to final media concentrations of $10^{-7}$ M FeCl$_3$, $4.8 \times 10^{-8}$ M MnCl$_2$, $4.0 \times 10^{-8}$ M CuSO$_4$, $10^{-7}$ M NiCl$_2$, and $10^{-8}$ M Na$_2$O$_3$Se in a $10^{-4}$ M ethylenediamine tetraacetic acid disodium salt (EDTA, Acros Organics, HV-88113-90) metal ion buffer system. All media amendments were sterile filtered through acid rinsed 0.2 µm filters before addition to final media, and final media equilibrated for at least 12 h before inoculation.

Established diatom cultures were first acclimated to low-metal media containing no additional Zn nor Co for at least three transfers before being used to inoculate biological duplicate cultures at 1% volume. Growth of all cultures was monitored by measuring in vivo fluorescence on a near-daily basis as a proxy for chlorophyll $a$ using a Turner TD-700 fluorometer, calibrated prior to measurement with a solid standard. Computed ratios of [Zn$^{2+}$] and [Co$^{2+}$] to total metal concentrations, the values of which are $10^{-3.99}$ and $10^{-3.63}$ respectively, were used to convert total added metal concentrations to free ion concentrations and are the same as those used by Sunda and Huntsman[2]. Other inorganic complexes of Zn and Co are also present in proportional abundance to the divalent cation form. The f/2 seawater media base used for all growth experiments was analyzed via inductively coupled plasma mass spectrometry (ICP-MS) to determine background media concentrations of total Zn and Co (0.9 nmol L$^{-1}$ and 0.1 nmol L$^{-1}$, respectively). Briefly, the f/2 media base was diluted 1:10 into 5% nitric acid containing 1 ppb indium solution before being analyzed in duplicate on a Thermo ICAP-Q plasma mass spectrometer calibrated to a multi-element standard curve (Spex Certiprep) over a range of 1–20 ppb ($R^2 > 0.9994$ for all metals). Samples were analyzed in KED mode after an 85 s sample uptake window and element mass windows were scanned 3 times during measurements. The 1 ppb indium internal standard was used to correct for variation in sample delivery and plasma suppression between samples. Process blank digestions containing acid but no media were performed in parallel, and blank values were subtracted from measured concentrations.

**Proteomic analysis of experiment cultures.** Biomass from duplicate 25 mL cultures was collected during log phase growth by centrifugation at $14,610 \times g$ for 40 min at 4 °C. Proteins were extracted, alkylated, and reduced using a modification of previously published methods[53,54]. The extract was resuspended in lysis buffer (50 mM HEPES, pH 8.5 (Thermo Scientific J63218.AK1% SDS (Thermo Scientific 28312) and heated for 10 min at 95 °C, then incubated for 30 min at room temperature with gentle shaking at 350 RPM. The extract was then centrifuged at $14,100 \times g$ for 20 min to remove cellular debris. The supernatant was transferred to an ethanol-washed microcentrifuge tube and 2 µL of benzonase nuclease (MilliporeSigma 707463) was added before incubating for 30 min at 37 °C. Samples were then alkylated by adding 5 µL of 200 mM dithiothreitol (DTT; Invitrogen D1532) in 50 mM HEPES pH 8.5 per 100 µL of lysate and incubated at 45 °C for 30 min. Ten µL of 400 mM iodoacetamide (Thermo Scientific 122270050) in 50 mM HEPES pH 8.5 was then added, followed by incubation in the dark for 30 min at 24 °C. 10 µL of 200 mM DTT in 50 mM HEPES pH 8.5 per 100 µL of lysate was then added to quench the reaction.

Proteins were isolated and washed using a magnetic bead method adapted from Hughes et al. (2014), using a 1:1 mixture of hydrophobic and hydrophilic Sera-Mag SpeedBeads (MilliporeSigma GE17152104010). Magnetic beads were added to the protein mixture at a concentration of 20 µg/100 µL of total reaction. The solution was acidified with 10% formic acid (Thermo Scientific 147932500) to pH 2. Acetonitrile (ACN, Thermo Scientific 043440.K7) was then immediately added to a final concentration of 50% and samples were incubated for 15 min at 37 °C, then for 30 min at room temperature before being placed on magnetic racks for 2 min. The mixture was then washed with an equal volume of 70% ethanol (2×) and acetonitrile (1×). After removing the final ACN wash, the proteins were reconstituted in 50 mM HEPES (pH 8.0) and trypsin (Thermo Scientific Pierce 90059) digested for 14 h at 37 °C. The mixture was resuspended, washed with ACN, and placed on the magnetic rack. Proteins adhered to beads while ACN was removed by pipetting. The peptide sample was reconstituted in 2% DMSO (Thermo Scientific J66650.AK) and acidified with 1% formic acid to bring to a final concentration of 0.1 µg/µL. Protein concentration was quantified using a colorimetric assay at 562 nm (Pierce BCA total protein assay, Thermo Scientific 23225).

Whole cell protein digests were analyzed in technical triplicate by liquid chromatography mass spectrometry (LC-MS) using a Paradigm MS4 HPLC system (Michrom) with reverse phase chromatography, a Michrom ADVANCE source, and a Thermo Scientific Q Exactive hybrid quadrupole-orbitrap mass spectrometer. Mass spectra were searched using Proteome Discoverer's SEQUEST HT algorithm (Thermo, Inc.) with a precursor-ion mass tolerance of 100 ppm and product-ion mass tolerance of 0.02 Da. *Chaetoceros* RS19 mass spectra were searched against a *Chaetoceros* RS19 translated transcriptome database that was generated under iron limitation and otherwise replete conditions. Database search results were processed and visualized using spectral counting abundance scoring within Scaffold 5.0 (Proteome Software, Inc.). Protein abundances are reported as total normalized spectra with the following parameters: 95.0% minimum peptide threshold, 99.0% minimum, and 2 peptides minimum protein threshold, 0.0% peptide false discovery rate (FDR), and 0.0% protein FDR.

Putative ZCRP homologs among eukaryotic oceanic taxa were identified by BLAST searching (using cutoff of $E = 5.0 \times 10^{-5}$) the *P. tricornutum* ZCRP-A and ZCRP-B protein sequences against all available transcriptomes in the Marine Microbial Eukaryotic Transcriptome Sequencing Project database (MMETSP;

https://www.imicrobe.us/#/projects/104)[46]. One putative homolog per species was selected for further data analysis. All proteins were aligned using the MUSCLE algorithm[55] with default parameters within MEGA11[56]. Phylogenetic trees of ZCRP-A and ZCRP-B proteins were constructed in MEGA11 using the UPGMA method[57] with 1000 repetitions of the bootstrap test[58].

**Knockout of ZCRP-A in P. tricornutum.** Primer sequences are provided in Supplementary Data 1. Axenic *P. tricornutum* CCMP632 cultures were grown in artificial seawater medium at 18 °C under constant cool white fluorescent lighting (50 µmol photon m$^{-2}$ s$^{-1}$). CRISPR-cas9 gene editing was used to knock out *ZCRP-A* in *P. tricornutum*[59]. Briefly, a target sequence $5' – \text{GN}^{20}\text{NGG} – 3'$ was found within the conserved GTPase domain of *ZCRP-A*. Single guide RNA nucleotides (sgRNAs) designed to target this sequence were created by annealing and extending primers to create a ~100 bp insert. sgRNAs were then inserted into a guide RNA (gRNA) vector plasmid containing the sRNAi promoter via two-piece Gibson assembly[60]. gRNA plasmids were propagated in chemically competent *Escherichia coli* (Epi300, Epicenter, WI, USA) cells grown on agar media supplemented with ampicillin (100 µg mL$^{-1}$), tetracycline (5 µg mL$^{-1}$), Zeocin (Invivo-Gen, catalog code ant-zn-05; 25 µg mL$^{-1}$), or combinations of these as needed. A separate donor plasmid containing the ShBle cassette conferring resistance to the selection antibiotic Zeocin was likewise propagated in *E. coli* as was the final donor plasmid containing Cas9 under control of the FcpB promoter (FcpB_Cas9). Colony PCR and gel electrophoresis were used to select successfully transformed colonies. sRNAi, FcpB_Cas9, and ShBle donor plasmids are delivered to *P. tricornutum* cells via particle bombardment. Bombarded *P. tricornutum* colonies were grown on agar supplemented with Zeocin (20 µg mL$^{-1}$). Colonies were sequenced and bi-allelic knockout of *ZCRP-A* was confirmed. Proteomic analysis comparing *P. tricornutum* ZCRP-A knockout and wild-type cell lines confirmed the nondetection of ZCRP-A in the knockout line (Supplementary Fig. 5).

**Overexpression of ZCRP-A and ZCRP-B in P. tricornutum.** The plasmids PT-YFP-41508 and PT-YFP-49168 were constructed to overexpress the ZCRP-A and ZCRP-B proteins in *P. tricornutum*. The YFP tag was included to determine the cellular localization of these proteins by fluorescent microscopy. Plasmids were constructed by Golden Gate assembly[61] using pUC19 backbone plasmids and contain an N-terminal YFP fused to the genes of interest in addition to an ShBle cassette conferring Zeocin resistance. Plasmid sequences were verified and plasmids were transformed into *E. coli* and mobilized into *P. tricornutum* by bacterial conjugation. *P. tricornutum* cells containing YFP plasmids were grown in liquid culture supplemented with Zeocin. Epifluorescent microscopy was performed using a Zeiss Axioscope to determine protein localization. Proteomic analysis comparing *P. tricornutum* ZCRP-A overexpression, ZCRP-B overexpression, and wild-type cell lines confirmed increased abundances of ZCRP-A and ZCRP-B in the respective overexpression lines compared to the wild-type (Supplementary Fig. 5).

**Radiotracer cobalt uptake.** The radiotracer $^{57}$Co as $^{57}$CoCl$_2$ (Isotope Products Laboratories, Valencia, CA; 99% radionuclide purity in 0.1 M HCl) was used to measure the rate of $^{57}$Co uptake in wild type, *ZCRPA* knockout, *ZCRPA* over-expression, and *ZCRP-B* overexpression lines of Zn and Co limited *P. tricornutum*. Biological triplicate cultures in mid-log phase were spiked with approximately 0.1 pM $^{57}$CoCl$_2$ ($6.3 \times 10^{-5}$ µCi/mL $^{57}$CoCl$_2$) and were incubated at room temperature under constant cool white fluorescent lighting (90 µmol photon m$^{-2}$ s$^{-1}$) for 24 h. All cultures were grown in natural 0.2 µm filtered surface South Pacific seawater (containing background concentrations of 7.7 pM Co and 4.0 nM Zn) that was unamended with the exception of 1 nM total added Fe (as FeCl$_3$) to prevent any potential effects arising from Fe limitation. 200 µL of each culture was collected as a measure of total $^{57}$Co activity. Remaining sample volumes were vacuum-filtered through 3µm polycarbonate filters (Versapore) and collected. $^{57}$Co radioactivity on each filter was measured using a Canberra Germanium Gamma detector. Counts per minute at 122 keV were corrected for decay and normalized to percent uptake per day, calculated by dividing the activity on each filter by the total $^{57}$Co activity. Percent uptake was further normalized to culture fluorescence (fsu) measured just before $^{57}$Co addition as a proxy for cell counts. Samples were counted long enough to achieve a counting error < 5%.

**Metaproteomic sampling.** Metaproteomic sampling methods for the METZYME cruise have been described previously[47]. Briefly, biomass collection for metaproteomics was performed using McLane pumps (McLane Research Laboratories) secured to a trace metal clean winch line. Membrane filters targeting specific size classes were sectioned for 'omics analyses. Metaproteomic identifications from the 3–51 µm size fraction, which contained eukaryotic protists and particle-associated bacteria, was searched for the *P. tricornutum* ZCRP-A and ZCRP-B sequences using BLAST+.

**Statistical analysis and data visualization.** To assess the significance of the relationship between metal concentration and ZCRP-A/B abundance, we utilized the Kendall-tau rank correlation test. This test does not assume a priori a relationship between two indices, and instead tests whether the ranked order of two quantities is significantly correlated. The correlation is two-tailed and can vary

from −1 to 1, with −1 being perfectly anti-correlated, 0 being no relationship at all, and 1 being perfectly correlated[62]. For the radiotracer uptake experiments, a one-way ANOVA and post-hoc Dunnet test (significance cutoff $p = 0.05$) were used to assess significant differences in the extent of Co uptake after 24 h among genetically modified *P. tricornutum* lines.

Protein section plots were created in Ocean Data View 4 v4.7.10 using Data-Interpolating Variational Analysis (DIVA) interpolation[63]. The TOPCONS webserver at https://topcons.cbr.su.se/ was used for prediction of membrane topology of *P. tricornutum* ZCRP-B[27,28]. Sequence alignments were visualized using BOXSHADE (http://www.ch.embnet.org/software/BOX_form.html). Bar plots and scatter plots were created in Excel 2019 or with ggplot2 v3.2.1.

## Data availability

Diatom proteins identified as ZCRPs are listed in Table S3 with their protein IDs. METZYME dissolved metal and pigment data is available through the NSF's Biological and Chemical Oceanography Data Management Office (BCO-DMO) repository under project number 2236. Growth rates for *T. pseudonana*, *P. tricornutum* and *P. delicatissima* are available under BCO-DMO dataset number 807316. The mass spectrometry global proteomics data for METZYME, the *Chaetoceros* RS19 translated transcriptome used in this study, and the diatom proteomes generated in this study have been deposited with the ProteomeXchange Consortium through the PRIDE[64] repository under dataset identifiers PXD014230, PXD026895, and PXD026953, respectively. Annotated proteomic databases used for *T. pseudonana* and *P. tricornutum* (Thaps3 and CCAP 1055/1 v2.0 Phatr2, all models) are available online from the Joint Genome Institute (https://mycocosm.jgi.doe.gov/Thaps3/Thaps3.info.html, https://mycocosm.jgi.doe.gov/Phatr2/Phatr2.home.html)[17,65]. *P. delicatissima* UNC1205 mass spectra were searched against a translated transcriptome available through the EukProt database (ID EP00533; https://figshare.com/articles/dataset/EukProt_a_database_of_genome-scale_predicted_proteins_across_the_diversity_of_eukaryotic_life/12417881/2)[66]. Putative ZCRP homologs were identified within transcriptomes in the Marine Microbial Eukaryotic Transcriptome Sequencing Project database (MMETSP; https://www.imicrobe.us/#/projects/104). ZCRP-A homologs were modeled using entry 4IXM.2.b within the Swiss Model repository (https://swissmodel.expasy.org/repository).

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

## Acknowledgements

We thank Adrian Marchetti and Tracy Mincer for cultures of *P. delicatissima* UNC1205 and *P. tricornutum* CCMP632. We thank Caitlyn Kelleher for feedback on an earlier draft. This work was funded by the National Science Foundation (OCE-1736599 and OCE-1657766), NIH (R01GM135709), Gordon and Betty Moore Foundation (GBMF3782) to M.A.S., and Simons Foundation award 544236 to N.R.C. This work was further supported by the National Science Foundation (NSF-OCE-1756884 and NSF-MCB-1818390), United States Department of Energy (DE-SC0018344), and Gordon and Betty Moore Foundation grants GBMF3828 and GBMF5006 to A.E.A.

## Author contributions

All authors contributed to data acquisition and analysis. R.M.K. and M.A.S. designed the experiments and R.M.K. conducted Zn/Co physiological experiments, proteomic extractions and analyses, sequence analyses, and analysis of metaproteomic data. M.A.M. and A.E.A. implemented reverse genetic techniques. R.M.K. and M.A.M. provided the microscopy images. N.R.C. contributed the dissolved Zn dataset. N.J.H. contributed the dissolved Co dataset. M.R.M. assisted with proteomic analyses. D.M.M. originally isolated *Chaetoceros* RS19 used in this study and assisted in the maintenance of cultures. G.R.D. contributed the pigment dataset. A.V.S. contributed to the discussion regarding the carbonic anhydrase data and Kendall correlation analyses. A.E.A. contributed the translated transcriptome for *Chaetoceros* RS19 used as a reference database for proteomic analyses. R.M.K. and M.A.S. wrote the manuscript. All authors commented on and approved the final submitted manuscript.

## Competing interests

The authors declare no competing interests.
