## [Peer Review File · Nature Communications]

Adaptive responses of marine diatoms to zinc scarcity and ecological implicationsREVIEWER COMMENTS

Reviewer #1 (Remarks to the Author):

This is a very well written manuscript describing an impressive study exploring the diatom (and other eukaryotic phytoplankton) responses to Zinc scarcity. The oceanographic community has been aware of the potential for Zn limitation in the oceans for years, but it wasn't until very recently that the combination of trace metal clean sampling, trace metal clean culturing, OMIC analyses, and genetic characterization have made this work even possible. It is exciting to see this all combined into one manuscript. I have a few comments for clarification that are mainly editorial.

In results/discussion, clarify that no Zn was added to the Co treatments (as no Co was added to the Zn treatments). It's in the methods, but those are at the end (and one read of the first sentence could imply that Co and Zn growth experiments were done separately – as in there was a Zn range and a Co range). It becomes clear as you read through that this is not the case but might as well state it up front.

Figure 1 – for clarity it would be best to use the same units for concentration in a/c and c/d.

Lines 83-86 probably should refer to figure 2 a-d or this sentence should be moved to after figure 2 is discussed because at this point this has only been shown in one diatom.

Line 180 – perhaps explain why that would be a reasonable assumption (i.e., because Zn is important for CAs and a ZCRP-A knockout may have reduced Zn acquisition capabilities). I realize that later the authors make clear that they don't expect a direct interaction between the ZCRPs and the CAs, but since Zn is linked to the CCMs, maybe state it explicitly here?

Line 340 – perhaps Fig. S6 should be in the main document? It seems important to the story.

Really minor formatting comments:

The manuscript goes back and forth on using [Co] and Co concentration. Double check that this is consistent. The supplemental figures are referenced out of order.

Line 47 “dZn”

Line 48 is “GS” short for something?

Line 54 “Zn-” to match with “Co-”

Reviewer #2 (Remarks to the Author):

In the manuscript “Adaptive Responses of Marine Diatoms to Zinc Scarcity and Ecological Implications”, Saito and co-workers seek to determine the adaptive response of four marine diatoms to zinc and cobalt deficiency, and secondly, to use these as biomarkers to identify oceanic regions where Zn is limited. Diatoms play a significant role in the marine environment by being major contributors to photosynthetic carbon fixation from oceans via the action of a Zn-metalloenzyme, carbonic anhydrase. The availability of Zn at the ocean surface is thought to be low and diatoms have evolved the ability to use Co or Cd as a Zn substitute in certain CAs. The authors use global proteomic analysis to identify two previously uncharacterised proteins expressed under zinc and cobalt limitation, renamed ZCRP-A and ZCRP-B. Homology modelling demonstrates identity to bacterial metal homeostasis proteins, although due to the genetic distance to the bacterial counterparts, the biological function of ZCRP-A and ZCRP-B cannot be easily predicted. The authors identify the cellular location of each protein and confirm their expression at the ocean surface.

An understanding of the physiological response of diatoms to zinc is certainly lacking and this is an interesting study which tackles this, however there are some areas which need to be addressed. Although worthy of publication, I am not convinced that the quantity and quality of data is sufficient for the current journal.

Comments to be addressed:

- 1) More effort should be made in the introduction to highlight the significance of the findings.
- 2) Line 82. Salmonella do not possess a CobW. B12 biosynthesis is via an anaerobic cobalt chelatase CbiK. Please clarify the homology. It is this reviewer's understanding that Salmonella (similar to E. coli) has two CobW-like proteins, YeiR and YjiA.
- 3) Figure 2. High affinity metal binding by COG0523 proteins requires association of the protein with magnesium and GTP which is thought to allow formation of a thiol rich metal binding site. It seems more likely to this reviewer that a Zn/Co site would utilise the three cysteine residues in the CXCC motif. Notably the 41XM.2.b structure was generated in the absence of nucleoside and magnesium. Can the authors elaborate on this. Are there sequence differences in homologues that might indicate Zn vs Co as the preferred metal.
- 4) What is known of the regulatory mechanism of ZCRP-A and ZCRP-B. Do these diatoms have metalloregulators such as the bacterial Zur?
- 5) E. coli NikA, to which ZCRP-B is 30.5 % identical, promotes Ni uptake by binding a 1:2 Ni:Histidine complex, rather than a hydrated Ni ion. Is it possible that ZCRP-B binds a metal complex for uptake with associated transport machinery?

Reviewer #3 (Remarks to the Author):

REVIEW: NCOMMS-21-45671

Adaptive Responses of Marine Diatoms to Zinc Scarcity and Ecological Implications [Kellogg et al.]

Analytical approach

The manuscript describes the appropriate use of several techniques including proteomics, genetics, bioinformatics, and fluorescence microscopy to identify and characterize two metal-responsive proteins (referred to as Zn/Co Responsive Proteins A and B) in four species of marine diatom, and to provide evidence of a role for these proteins in adapting to Zn scarcity in the open ocean.

Clarity and context

The manuscript, which is generally well-written, follows a logical progression from proteomic analysis of diatom cultures containing different concentrations of the divalent micronutrients zinc (Zn²⁺) and cobalt (Co²⁺) through structural and functional characterization of proteins ZCRP-A and ZCRP-B via sequence alignment, intracellular localization, and morphological analysis of Wt and mutant lines to comparisons of protein abundance with Zn²⁺, Co²⁺, and phytoplankton pigment concentrations in the surface waters of the Pacific Ocean.

Key results

The results provide compelling evidence that these two differentially-expressed proteins are involved in a metabolic response to Zn/Co limitation in diatoms (which, together with other phytoplankton, play a key role in the marine food web and biological carbon pump) and that Zn nutritional stress in the surface ocean could be more widespread than previously thought.

The suggestion that a component of the bacterial Ni transport complex Opp1 may have been recruited and repurposed for Zn acquisition as ZCRP-B (with which it shares over 25% identity) during the evolution of marine diatoms is one example of the novel insights provided by this comprehensive, multidisciplinary study.

Significance

The findings, which are supported by the results (see below), are significant in terms of understanding the role of micronutrient trace metals in marine primary production and microbial ecology (including harmful algal species) and, as such, warrant consideration for publication in Nature Communications.

Before receiving further consideration, however, the manuscript would benefit from some additional attention with regard to organization, presentation, and interpretation of the data. In particular, the authors are asked to consider the following.

Data and methodology

p.4 ll.75-78 Global proteomic analysis. Although the LC-MS/MS procedure used to identify proteins in tryptic digests is clearly described in the methods section (p.21, ll. 464-469), along with the other techniques used, no specific information is provided with regard to the confidence with which the two differentially-expressed Zn/Co-responsive proteins are first identified in *T. pseudonana* cultures. A list of the tryptic peptides matched to each protein sequence in the Thaps3 database and/or a figure (perhaps similar to the one used to show protein alignment in Fig. 5S) illustrating sequence coverage, along with a measure of confidence in the matches obtained, would seem appropriate given the importance of this initial step.

Validity

p.4 ll.88-92 Protein spectral abundance vs. metal concentration in cultures. According to Fig. 2a, ZCRP-A abundance is indeed inversely related to metal concentration in *T. pseudonana*. However, spectral abundance appears to increase with increasing metal concentration for both proteins between the lowest and next highest concentrations of Co^{2+} and Zn^{2+} in *P. tricornutum* and *P. delicatissima* (Fig. 2b and 2c). Perhaps the lowest concentrations of these two essential micronutrients caused a reduction in overall metabolism, including protein synthesis. In any case, this apparent deviation from what is otherwise a similar pattern to that observed in *T. pseudonana* should be addressed in the discussion.

References

p.10 ll.223-249 Mechanism of cellular Zn uptake. The suggested involvement of putative membrane-associated Zn-binding protein ZCRP-B in cellular uptake is intriguing, particularly the idea of the surface-tethered (or excreted) protein competing Zn away from natural ligands. Aristilde et al. (2012) describe a scenario in which weak natural Zn-binding ligands enhance cellular uptake of Zn in the coastal diatom *Thalassiosira weissflogii* via formation of transient ternary complexes with uptake molecules. The effect is most dramatic with Zn-limited cells, for which high-affinity transporters should be most effective at extracting Zn from natural ligands. This paper may be useful to the authors in further developing the concept of such a mechanism involving ZCRP-B. Recent work by Wong et al. (2021) and Nixon et al. (2021) involving the analysis of Cu ligands in Pacific surface waters (where Zn tends to be scarce) confirms the presence of organic ligands with the potential to regulate Zn uptake in these waters.

Aristilde, L. et al. Weak organic ligands enhance zinc uptake in marine phytoplankton. *Environ. Sci. Technol.* 46, 5438-5445 (2012).

Nixon, R.L. et al. Evidence for the production of copper-complexing ligands by marine phytoplankton in the subarctic northeast Pacific. *Mar. Chem.* 237, 104034 (2021).

Wong, H.W. et al. New insights into the biogeochemical cycling of copper in the subarctic Pacific: distributions, size fractionation, and organic complexation. *Limnol. Oceanogr.* 66, 1424-1439 (2021).

Suggested improvements

p.8 ll.169-181 Frustule morphology. Suggest that this paragraph be restructured so that the section beginning "Phenotypic plasticity..." (l. 174) and ending with "...supports this hypothesis" (l.179) is cut and pasted just after the first sentence (l.169). In this way the nature and significance of phenotypic plasticity in *P. tricornutum* are introduced before the results, which helps the flow of the discussion. Regarding the hypothesis that adopting the triradiate form could be a response to a disruption in carbon metabolism (l.179-180) it is possible that the global proteomic results may provide additional evidence for this (for example, with regard to CA expression).

Figure 4 Localization, topography, and Co uptake in *P. tricornutum*. This figure appears crowded and, as a result, some of the details are difficult to make out. Suggest that the micrographs (a) to (h) be presented in one figure and the remaining panels in another. Note that the composite micrograph showing localization of ZCRP-B to the cell membrane is in the panel labeled (e), not (b) as stated in the figure caption. This could be addressed by re-labeling the panels such that side-by-side comparisons of composite, DIC, YFP and Chl auto images are referenced as sequential pairs (a, b), (c,d), (e,f) and (g,h), respectively, in the caption. This could be accommodated in the text by changing (Fig. 4a-d, Fig. S3a) (p.7 l.139) to (Fig. 4a, Fig. S3a) since the composite image (Fig. 4a) is sufficient to demonstrate localization of ZCRP-A. Incorporating panels (i) to (k) into a new figure also works in terms of the sequence in which they are referenced in the text, although this would necessitate re-numbering of the remaining figures.

p.11 ll.237-240 Abundance of ZIP proteins. The text refers to Fig. 5b regarding the relative abundance of ZCRP-B and ZIPs. However, Figure 5 shows the change in abundance of alpha and iota CAs with metal concentration (along with ZCRP-A and ZCRP-B) whereas the spectral abundance of ZIP 42755 in *P. tricornutum* is found in Supplementary Fig. S8b (and that of ZCRP-B in Fig. S8a).

p.15 ll.332-335 Correlation of protein abundance and metal concentration. The term 'zincocline' (l.334) is unfamiliar and, presumably, refers (by analogy with thermocline and halocline) to a fairly narrow depth range within the water column defined by a steep gradient in Zn concentration. The authors should define this term, annotate Fig. 6 accordingly, and/or re-word the sentence in which it appears to indicate a deepening of the upper layer in which dissolved Zn (and Co) are depleted. Since Fig. S7 appears to show the same information as Fig. 6f & g (albeit with a log vertical scale) consideration should be given to omitting the latter (and possibly including one or more panels from Fig. S6 instead).

The following edits are also suggested in order to help clarify the text:

P.2 ll.35-36 "Among the micronutrients required for diatom metabolism, zinc (Zn) is known to be particularly important".

p.2 l.44 replace "implicate" with "points to".

p.6 l.133 replace "imply" with "mean".

p.6 l.134-135 "...cellular function since the C-terminal domain, which is thought to be an indicator of protein-protein interaction specificity, is quite different in these two proteins".

p.7 l.146 "To date, connections between COG0523 proteins and utilization of Zn and Co have been explored primarily in prokaryotic organisms".

p.7 ll.152-153 delete "of life".

p.7 ll.153-155 "Having been the first to report the presence of Zn responsive COG0523 proteins in marine diatoms, we here present additional evidence in the form of genetic, physiological, protein localization and field data."

p.8 l.161 "...organisms in which they are found".

p.8 l.163 replace "yet" with "whereas".

p. l.166 "..., and that further metal binding and affinity assays may confirm..."

p.9 l. 201 "...implying that their function and regulation are independent of ZCRP-B".

p.10 ll. 216-219 "...fits the description of a high-affinity Zn uptake system observed in marine algae that is known to be induced at low free Zn²⁺ concentrations, suggesting that..."

p.13 ll.286-287 replace "lend further evidence to" with "provide further evidence for".

p.15 l.329 replace "dZn" with "Dissolved Zn".

p.15 l.331 replace "dCo" with "Dissolved Co".

p.16 l.350 replace "implicates" with "implies".

p.16 l.355 replace "...Zn contamination. Yet..." with "...Zn contamination, yet..."

p.16 l.356 replace "...surface ocean (e.g., Middag et al., 2019), and given..." with "...surface ocean (e.g., Middag et al., 2019). Given..."

p.16 l.361 "...concentrations of ~1 pM..."

p.16 l.363 replace "..., similar to how the..." with "..., just as..."

p.16 l.364 replace "have" with "has".

p.16 l.367 "...may, therefore, be..."

p.17 l.371 "These previously uncharacterized proteins are..."

p.17 l.374 replace "during" with "under".

p.17 ll.376 substitute "with" for "but".

p.17 l.379 substitute "also" for "furthermore".

p.18 l.391 "...with constant light (40 $\mu\text{mol photon m}^{-2} \text{s}^{-1}$).

p.18 l.393 "Thalassiosira pseudonana CCMP1335..."

p.18 l.400 "techniques".

p.18 l.403 "Milli-Q".

p.18 l.408 omit "that used by".

p.19 l.415 substitute "in" for "within".

p.19 l.424 "total metal concentrations".

p.19 l.430 "...nitric acid containing 1 ppb indium before being analyzed..."

p.20 ll. 441 and 443 substitute "The extract" for "Biomass".

p.20 ll. 448 and 449 substitute "Ten μL " for "10 μL ".

P.21 ll.468-469 "...with a precursor-ion mass tolerance of 100 ppm and product-ion mass tolerance of 0.02 Da".

p.21 l.479 New paragraph after "...protein FDR.?"

POINT BY POINT RESPONSE TO REVIEWERS

“Adaptive Responses of Marine Diatoms to Zinc Scarcity and Ecological Implications”
by Riss M. Kellogg, Mark A. Moosburner, Natalie R. Cohen, Nicholas J. Hawco, Matthew R. McIlvin, Dawn M. Moran, Giacomo R. DiTullio, Adam V. Subhas, Andrew E. Allen, and Mak A. Saito.

(Author responses in blue)

We thank all reviewers for their constructive feedback regarding our manuscript— with the exception of some minor wording choices, we have made all suggested changes.

Reviewer #1 (Remarks to the Author):

This is a very well written manuscript describing an impressive study exploring the diatom (and other eukaryotic phytoplankton) responses to Zinc scarcity. The oceanographic community has been aware of the potential for Zn limitation in the oceans for years, but it wasn't until very recently that the combination of trace metal clean sampling, trace metal clean culturing, OMIC analyses, and genetic characterization have made this work even possible. It is exciting to see this all combined into one manuscript. I have a few comments for clarification that are mainly editorial.

- Thank you for your appreciation of the study and constructive feedback.

In results/discussion, clarify that no Zn was added to the Co treatments (as no Co was added to the Zn treatments). It's in the methods, but those are at the end (and one read of the first sentence could imply that Co and Zn growth experiments were done separately – as in there was a Zn range and a Co range). It becomes clear as you read through that this is not the case but might as well state it up front.

- We have changed the first sentence of results/discussion to “Zn and Co growth rate experiments in which Zn or Co (omitting the other) were added to the growth media were conducted and harvested for proteomic analysis.”

Figure 1 – for clarity it would be best to use the same units for concentration in a/c and c/d.

- We have replaced the units for (c) and (d) with log[Zn] and log[Co] as suggested

Lines 83-86 probably should refer to figure 2 a-d or this sentence should be moved to after figure 2 is discussed because at this point this has only been shown in one diatom.

- Added a reference to Fig 2a-d, line 85 now reads “Based on their clear response to Zn and Co in the proteomes of multiple diatom species (Fig. 2a-d)...”

Line 180 – perhaps explain why that would be a reasonable assumption (i.e., because Zn is important for CAs and a ZCRP-A knockout may have reduced Zn acquisition capabilities). I realize that later the authors make clear that they don't expect a direct interaction between the ZCRPs and the CAs, but since Zn is linked to the CCMs, maybe state it explicitly here?

- Changed sentence to “As Zn^{2+} is the predominant metal cofactor used in diatom CAs, the adoption of the triradiate form in ZCRP-A knockout *P. tricornutum* cells may be a response to a disruption of the carbon concentrating mechanism caused by a reduction in Zn acquisition capability due to ZCRP-A knockout.

Line 340 – perhaps Fig. S6 should be in the main document? It seems important to the story.

- We have moved this figure (showing phytoplankton pigments over the METZYME cruise) to the main text, it is now Fig. 7.

Really minor formatting comments:

The manuscript goes back and forth on using [Co] and Co concentration. Double check that this is consistent. The supplemental figures are referenced out of order.

- Where appropriate grammatically, we have changed instances of “Zn concentration” and “Co concentration” to “[Zn^{2+}]” or “[Co^{2+}]”.
- We have now put the supplemental figures in the order they are referenced in the text.

Line 47 “dZn”

- Changed “total dissolved Zn” to “total dZn”

Line 48 is “GS” short for something?

- We believe GS stands for “GEOTRACES Surface”; there are two GEOTRACES reference standards for Zn, GD (D=deep) and GS (S=surface).
- Changed this description in line 48 to “GEOTRACES Zn surface water reference standard GS”

Line 54 “Zn-“ to match with “Co-“

- Changed “Zn” to “Zn-”

Reviewer #2 (Remarks to the Author):

In the manuscript “Adaptive Responses of Marine Diatoms to Zinc Scarcity and Ecological Implications”, Saito and co-workers seek to determine the adaptive response of four marine diatoms to zinc and cobalt deficiency, and secondly, to use these as biomarkers to identify

oceanic regions where Zn is limited. Diatoms play a significant role in the marine environment by being major contributors to photosynthetic carbon fixation from oceans via the action of a Zn-metalloenzyme, carbonic anhydrase. The availability of Zn at the ocean surface is thought to be low and diatoms have evolved the ability to use Co or Cd as a Zn substitute in certain CAs. The authors use global proteomic analysis to identify two previously uncharacterized proteins expressed under zinc and cobalt limitation, renamed ZCRP-A and ZCRP-B. Homology modelling demonstrates identity to bacterial metal homeostasis proteins, although due to the genetic distance to the bacterial counterparts, the biological function of ZCRP-A and ZCRP-B cannot be easily predicted. The authors identify the cellular location of each protein and confirm their expression at the ocean surface. An understanding of the physiological response of diatoms to zinc is certainly lacking and this is an interesting study which tackles this, however there are some areas which need to be addressed. Although worthy of publication, I am not convinced that the quantity and quality of data is sufficient for the current journal.

- Thank you for your constructive feedback. We appreciate the question of worthiness for the current journal—the potential for Zn limitation in the oceans has long been overlooked and we believe this study provides diverse data types to contribute substantial new knowledge in addressing this.

Comments to be addressed:

1) More effort should be made in the introduction to highlight the significance of the findings.

- We have changed the last introduction paragraph to the following: “In this study, we identified two Zn- and Co-responsive proteins in four distinct diatom species using a global proteomic approach and explored the abundance patterns of these proteins as a function of Zn²⁺ and Co²⁺ media concentrations. We employed reverse genetic techniques to further characterize these proteins and their putative functions. Coupling our laboratory findings to the field, we then present metaproteomic data demonstrating the presence of these proteins in surface Pacific waters. The presence of these proteins in the field implicates Zn stress within natural phytoplankton communities. Overall, this multidisciplinary study demonstrates that diatoms have a complex adaptive response to Zn scarcity, and that this scarcity may be a more important influence on primary productivity than previously recognized.”

2) Line 82. Salmonella do not possess a CobW. B12 biosynthesis is via an anaerobic cobalt chelatase CbiK. Please clarify the homology. It is this reviewer’s understanding that Salmonella (similar to E. coli) has two CobW-like proteins, YeiR and YjiA.

- Thank you for catching this—we had used the sequence with GenBank ID RFY65073.1 (found here in NCBI, <https://www.ncbi.nlm.nih.gov/protein/RFY65073.1?report=fasta>) annotated as “cobalamin biosynthesis protein CobW [Salmonella enterica]” (a poor annotation then)

- For this sentence, we will instead compare ZCRPA to the sequence of cobW in *Pseudomonas denitrificans*, which has a true cobW (https://www.ncbi.nlm.nih.gov/protein/WP_151189276.1)
- Changed sentence to “BLAST sequence alignments showed these proteins to be homologous with CobW-like proteins (with 31.69% identity relative to *Pseudomonas denitrificans* CobW) and with the nickel transport protein NikA (with 30.5% identity relative to *E. coli* NikA), respectively.

3) Figure 2. High affinity metal binding by COG0523 proteins requires association of the protein with magnesium and GTP which is thought to allow formation of a thiol rich metal binding site. It seems more likely to this reviewer that a Zn/Co site would utilize the three cysteine residues in the CXCC motif. Notably the 41XM.2.b structure was generated in the absence of nucleoside and magnesium. Can the authors elaborate on this. Are there sequence differences in homologues that might indicate Zn vs Co as the preferred metal.

- Indeed, work by Young et al. 2021 shows that binding of MgGTP by bacterial CobW results in a structural change resulting in a high-affinity site, and they also suggest that the Co binding site in CobW involves thiols likely derived from the CxCC motif and a tetrahedral geometry. We agree that it appears likely that the CxCC motif is key to metal binding. We utilized the 41XM.2.b model of *E. coli* YjiA in SWISS PROT to look at predicted Zn binding sites, as this is the crystal structure of Zn(II)-bound YjiA. The group that deposited this *E. coli* model, Sydor et al. (2013), comment in their corresponding manuscript that “Unfortunately, we have not been able to generate nucleotide-bound structures of YjiA itself,” thus a model of YjiA bound with Mg/nucleoside is unavailable for our comparisons. We use the models in Fig. 3 to demonstrate the presence of the CxCC putative Zn binding site in each diatom ZCRP-A homolog, but indeed the true protein conformation is likely that of the Mg/nucleoside-bound form as seen in bacteria.
- We have added the following text to address this: “Furthermore, previous work with bacterial CobW has shown that high affinity metal binding requires the association of the protein with magnesium (Mg²⁺) and GTP, with the thiol-containing Co²⁺ binding site in Mg²⁺GTP-CobW thought to be derived from the CxCC motif and a tetrahedral geometry.¹⁹ While no model of a Mg²⁺GTP-bound COG0523 protein yet exists, the presence of GTP-binding domains in diatom ZCRP-A (Fig. 3a) and a Zn²⁺ coordination environment comparable to that of YjiA suggests that high affinity metal binding in diatom ZCRP-A also likely requires association with Mg²⁺ and GTP.”
- Regarding if there are sequence differences in homologs that might indicate the preferred metal, a recent study of 80,000 protein sequences related to a COG0523 protein in *A. baumannii* by Edmonds et al. showed that proteins fell into distinct sequence similarity network clusters, with some ascribed to certain metal cofactors. Future studies could characterize metal preferences in diatom ZCRPA and how they are similar to or distinct from other COG0523 proteins.
- Edmonds, K. A., M. R. Jordan, and D. P. Giedroc. 2021. COG0523 proteins: a functionally diverse family of transition metal-regulated G3E P-loop GTP hydrolases from bacteria to man. *Metallomics* **13**. doi:10.1093/mtomcs/mfab046

- Sydor, A. M., M. Jost, K. S. Ryan, K. E. Turo, C. D. Douglas, C. L. Drennan, and D. B. Zamble. 2013. Metal Binding Properties of Escherichia coli YjiA, a Member of the Metal Homeostasis-Associated COG0523 Family of GTPases. *Biochemistry* **52**: 1788–1801. doi:10.1021/bi301600z
- Young, T. R., M. A. Martini, A. W. Foster, and others. 2021. Calculating metalation in cells reveals CobW acquires CoII for vitamin B12 biosynthesis while related proteins prefer ZnII. *Nat. Commun.* **12**: 1195. doi:10.1038/s41467-021-21479-8

4) What is known of the regulatory mechanism of ZCRP-A and ZCRP-B. Do these diatoms have metalloregulators such as the bacterial Zur?

- The Zn sensors and regulatory factors responsible for regulating the expression of ZCRPs in response to low Zn are yet unknown, and to our knowledge, there is no Zur homolog equivalent in these marine diatoms. A BLASTp search of *E. coli* Zur against all available protein databases of *T. pseudonana* and *P. tricornutum* (JGI) turns up no significant hits.

5) *E. coli* NikA, to which ZCRP-B is 30.5 % identical, promotes Ni uptake by binding a 1:2 Ni:Histidine complex, rather than a hydrated Ni ion. Is it possible that ZCRP-B binds a metal complex for uptake with associated transport machinery?

- Very possible, as also pointed out by reviewer 3 (see below). We have incorporated some text into the discussion of ZCRPB regarding a previous study by Aristilde et al. 2012: “Aristilde and colleagues have previously demonstrated that weak natural Zn-binding ligands containing cysteine do indeed enhance cellular Zn uptake within the diatom *Thalassiosira weissflogii*, with heightened effects in Zn-limited compared to Zn-replete cells.³⁹ They proposed the formation of a transient tertiary complex between the Zn-bound ligand and Zn transporters (ZIPs and heavy metal P-type ATPases) at the cell surface, which could be mediated by a surface-tethered Zn binding ligand such as ZCRP-B.”

Reviewer #3 (Remarks to the Author):

REVIEW: NCOMMS-21-45671

Adaptive Responses of Marine Diatoms to Zinc Scarcity and Ecological Implications [Kellogg et al.]

Analytical approach

The manuscript describes the appropriate use of several techniques including proteomics, genetics, bioinformatics, and fluorescence microscopy to identify and characterize two metal-responsive proteins (referred to as Zn/Co Responsive Proteins A and B) in four species of marine diatom, and to provide evidence of a role for these proteins in adapting to Zn scarcity in the open ocean.

Clarity and context

The manuscript, which is generally well-written, follows a logical progression from proteomic analysis of diatom cultures containing different concentrations of the divalent micronutrients zinc (Zn^{2+}) and cobalt (Co^{2+}) through structural and functional characterization of proteins ZCRP-A and ZCRP-B via sequence alignment, intracellular localization, and morphological analysis of Wt and mutant lines to comparisons of protein abundance with Zn^{2+} , Co^{2+} , and phytoplankton pigment concentrations in the surface waters of the Pacific Ocean.

Key results

The results provide compelling evidence that these two differentially-expressed proteins are involved in a metabolic response to Zn/Co limitation in diatoms (which, together with other phytoplankton, play a key role in the marine food web and biological carbon pump) and that Zn nutritional stress in the surface ocean could be more widespread than previously thought.

The suggestion that a component of the bacterial Ni transport complex Opp1 may have been recruited and repurposed for Zn acquisition as ZCRP-B (with which it shares over 25% identity) during the evolution of marine diatoms is one example of the novel insights provided by this comprehensive, multidisciplinary study.

Significance

The findings, which are supported by the results (see below), are significant in terms of understanding the role of micronutrient trace metals in marine primary production and microbial ecology (including harmful algal species) and, as such, warrant consideration for publication in Nature Communications.

Before receiving further consideration, however, the manuscript would benefit from some additional attention with regard to organization, presentation, and interpretation of the data. In particular, the authors are asked to consider the following.

- Thank you for your constructive feedback and careful attention to detail.

Data and methodology

p.4 ll.75-78 Global proteomic analysis. Although the LC-MS/MS procedure used to identify proteins in tryptic digests is clearly described in the methods section (p.21, ll. 464-469), along with the other techniques used, no specific information is provided with regard to the confidence with which the two differentially-expressed Zn/Co-responsive proteins are first identified in *T. pseudonana* cultures. A list of the tryptic peptides matched to each protein sequence in the Thaps3 database and/or a figure (perhaps similar to the one used to show protein alignment in

Fig. 5S) illustrating sequence coverage, along with a measure of confidence in the matches obtained, would seem appropriate given the importance of this initial step.

- We have created a new supplemental figure to display this information:

a. *T. pseudonana* ZCRP-A, $\log[\text{Zn}^{2+}] = -11.95 \text{ M}$

Jgi|Thaps3|3054|fgenes1_pg.C_chr_2001009 (100%), 43,779.4 Da
 Jgi|Thaps3|3054|fgenes1_pg.C_chr_2001009
 12 exclusive unique peptides, 14 exclusive unique spectra, 40 total spectra, 166/391 amino acids (42% coverage)

b. *T. pseudonana* ZCRP-A, $\log[\text{Co}^{2+}] = -11.63 \text{ M}$

Jgi|Thaps3|3054|fgenes1_pg.C_chr_2001009 (100%), 43,779.4 Da
 Jgi|Thaps3|3054|fgenes1_pg.C_chr_2001009
 9 exclusive unique peptides, 10 exclusive unique spectra, 26 total spectra, 134/391 amino acids (34% coverage)

c. *T. pseudonana* ZCRP-B, $\log[\text{Zn}^{2+}] = -11.95 \text{ M}$

fgenes1_pg.C_bd_23x33000033 (100%), 68,819.6 Da
 Jgi|Thaps3_bd|938|fgenes1_pg.C_bd_23x33000033
 18 exclusive unique peptides, 20 exclusive unique spectra, 165 total spectra, 226/624 amino acids (36% coverage)

d. *T. pseudonana* ZCRP-B, $\log[\text{Co}^{2+}] = -11.63 \text{ M}$

fgenes1_pg.C_bd_23x33000033 (100%), 68,819.6 Da
 Jgi|Thaps3_bd|938|fgenes1_pg.C_bd_23x33000033
 21 exclusive unique peptides, 30 exclusive unique spectra, 167 total spectra, 291/624 amino acids (47% coverage)

- Figure S1.** Exclusive unique peptides identified for *Thalassiosira pseudonana* CCMP1335 ZCRP-A (a,b) and ZCRP-B (c,d) in low Zn^{2+} ($\log[\text{Zn}^{2+}] = -11.95 \text{ M}$) and low Co^{2+} ($\log[\text{Co}^{2+}] = -11.63 \text{ M}$) treatments visualized in Scaffold 5. The number of exclusive unique peptides that map to each protein sequence are as follows: (a) 12, (b) 9, (c) 18, and (d) 21. For both proteins in all cases, protein probabilities were 100%.
- We have changed the text to the following: “These proteins were annotated as “CobW/HypB/UreG, nucleotide binding domain” and “Bacterial extra-cellular solute binding domains”, respectively, within the manually curated JGI Thaps3 *T. pseudonana* genome 17 and were identified in *T. pseudonana* cultures with high confidence (≥ 9 exclusive unique peptides, 100% protein probability; Fig. S1).

Validity

p.4 Il.88-92 Protein spectral abundance vs. metal concentration in cultures. According to Fig. 2a, ZCRP-A abundance is indeed inversely related to metal concentration in *T. pseudonana*. However, spectral abundance appears to increase with increasing metal concentration for both proteins between the lowest and next highest concentrations of Co^{2+} and Zn^{2+} in *P. tricornutum* and *P. delicatissima* (Fig. 2b and 2c). Perhaps the lowest concentrations of these two essential

micronutrients caused a reduction in overall metabolism, including protein synthesis. In any case, this apparent deviation from what is otherwise a similar pattern to that observed in *T. pseudonana* should be addressed in the discussion.

- It is true that ZCRP spectral counts in *P. delicatissima* and *P. tricornutum* increase in abundance between the lowest and next highest Co and Zn concentrations— the reviewer brings up a good point that this could be due to increased cellular stress in those two diatom species at the lowest added metal concentrations.
- We have added the sentence “unlike the continuous decrease in ZCRP abundances with increasing metal concentrations observed in *T. pseudonana* and *Chaetoceros* RS19, ZCRP abundances in *P. tricornutum* and *P. delicatissima* at the lowest metal treatments were somewhat smaller compared to their abundances in the next highest metal treatments, possibly owing to increased cellular stress and reduction in overall metabolism at the lowest added metal treatment in these diatoms” to the discussion.

References

p.10 ll.223-249 Mechanism of cellular Zn uptake. The suggested involvement of putative membrane-associated Zn-binding protein ZCRP-B in cellular uptake is intriguing, particularly the idea of the surface-tethered (or excreted) protein competing Zn away from natural ligands. Aristilde et al. (2012) describe a scenario in which weak natural Zn-binding ligands enhance cellular uptake of Zn in the coastal diatom *Thalassiosira weissflogii* via formation of transient ternary complexes with uptake molecules. The effect is most dramatic with Zn-limited cells, for which high-affinity transporters should be most effective at extracting Zn from natural ligands. This paper may be useful to the authors in further developing the concept of such a mechanism involving ZCRP-B. Recent work by Wong et al. (2021) and Nixon et al. (2021) involving the analysis of Cu ligands in Pacific surface waters (where Zn tends to be scarce) confirms the presence of organic ligands with the potential to regulate Zn uptake in these waters.

Aristilde, L. et al. Weak organic ligands enhance zinc uptake in marine phytoplankton. *Environ. Sci. Technol.* 46, 5438-5445 (2012).

Nixon, R.L. et al. Evidence for the production of copper-complexing ligands by marine phytoplankton in the subarctic northeast Pacific. *Mar. Chem.* 237, 104034 (2021).

Wong, H.W. et al. New insights into the biogeochemical cycling of copper in the subarctic Pacific: distributions, size fractionation, and organic complexation. *Limnol. Oceanogr.* 66, 1424-1439 (2021).

- Thank you for bringing these studies to our attention— our hypothesized mechanism of action regarding Zn uptake involving ZCRP-B is indeed reminiscent of the proposed mechanism in Aristilde 2012 involving the creation of transient ternary complexes. To the section of discussion of ZCRP-B, we have added the following:
- “Aristilde and colleagues have previously demonstrated that weak natural Zn-binding ligands containing cysteine do indeed enhance cellular Zn uptake within the diatom

Thalassiosira weissflogii, with heightened effects in Zn-limited compared to Zn-replete cells.³⁹ They proposed the formation of a transient tertiary complex between the Zn-bound ligand and Zn transporters (ZIPs and heavy metal P-type ATPases) at the cell surface, which could be mediated by a surface-tethered Zn binding ligand such as ZCRP-B.”

Suggested improvements

p.8 ll.169-181 Frustule morphology. Suggest that this paragraph be restructured so that the section beginning “Phenotypic plasticity...” (l. 174) and ending with “...supports this hypothesis” (l.179) is cut and pasted just after the first sentence (l.169). In this way the nature and significance of phenotypic plasticity in *P. tricornutum* are introduced before the results, which helps the flow of the discussion. Regarding the hypothesis that adopting the triradiate form could be a response to a disruption in carbon metabolism (l.179-180) it is possible that the global proteomic results may provide additional evidence for this (for example, with regard to CA expression).

- We have moved the sentence “Phenotypic plasticity in *P. tricornutum* is well documented. Two basic cell morphotypes, fusiform and triradiate, are found in natural liquid environments. It is thought that by adopting the triradiate form, a cell increases its surface area and thus the area of membrane available for enzymatic activity or molecular diffusion of dissolved inorganic carbon (DIC) into the cell. The triradiate form is known to be more common under DIC limiting conditions, which supports this hypothesis²⁷” to the very beginning of the paragraph to achieve better flow.
- We agree that the proteomic data comparing KO to WT PT lines is useful here— the relative increase in MnCA in the KO compared to the WT (Fig. S2) could imply that due to ZCRPA knockout, the diatom suffers a reduced Zn-binding capacity and thus responds by upregulating the Mn-utilizing CA.
- We have added the sentence “This is consistent with the observed relative increase in Mn²⁺-utilizing CA (t-CA) in the knockout line compared to the wild-type (Fig. S2).” We will refrain from going into further detail in this paragraph, since MnCA is formally introduced later in the text.

Figure 4 Localization, topography, and Co uptake in *P. tricornutum*. This figure appears crowded and, as a result, some of the details are difficult to make out. Suggest that the micrographs (a) to (h) be presented in one figure and the remaining panels in another. Note that the composite micrograph showing localization of ZCRP-B to the cell membrane is in the panel labeled (e), not (b) as stated in the figure caption. This could be addressed by re-labeling the panels such that side-by-side comparisons of composite, DIC, YFP and Chl auto images are referenced as sequential pairs (a, b), (c,d), (e,f) and (g,h), respectively, in the caption. This could be accommodated in the text by changing (Fig. 4a-d, Fig. S3a) (p.7 l.139) to (Fig. 4a, Fig. S3a) since the composite image (Fig. 4a) is sufficient to demonstrate localization of ZCRP-A. Incorporating panels (i) to (k) into a new figure also works in terms of the sequence in which

they are referenced in the text, although this would necessitate re-numbering of the remaining figures.

- We have relabeled the panels a-h as suggested, the caption for Fig. 4 now reads “...showing localization of (a) ZCRP-A to the chloroplasts and (b) ZCRP-B to the cell membrane...Composite images are stacks of the individual channels (c,d) differential interference contrast (DIC), (e,f) yellow fluorescent protein (YFP), and (g,h) chlorophyll autofluorescence (Chl auto).
- Changed (Fig. 4a-d, Fig. S3a) to (Fig. 4a, Fig. S3a) as suggested.
- We would prefer to keep this data within one figure rather than split it up, in order to keep the predicted ZCRPB transmembrane domain plot (j) close to the YFP ZCRPB image in (b) showing confirmed localization to the membrane.

p.11 ll.237-240 Abundance of ZIP proteins. The text refers to Fig. 5b regarding the relative abundance of ZCRP-B and ZIPs. However, Figure 5 shows the change in abundance of alpha and iota CAs with metal concentration (along with ZCRP-A and ZCRP-B) whereas the spectral abundance of ZIP 42755 in *P. tricornutum* is found in Supplementary Fig. S8b (and that of ZCRP-B in Fig. S8a).

- Thank you for catching this, the figure reference has been corrected.

p.15 ll.332-335 Correlation of protein abundance and metal concentration. The term ‘zincocline’ (l.334) is unfamiliar and, presumably, refers (by analogy with thermocline and halocline) to a fairly narrow depth range within the water column defined by a steep gradient in Zn concentration. The authors should define this term, annotate Fig. 6 accordingly, and/or re-word the sentence in which it appears to indicate a deepening of the upper layer in which dissolved Zn (and Co) are depleted. Since Fig. S7 appears to show the same information as Fig. 6f & g (albeit with a log vertical scale) consideration should be given to omitting the latter (and possibly including one or more panels from Fig. S6 instead).

- We agree that the use of the word “zincocline” may not be intuitive and have changed the text accordingly, the sentence now reads “This was coincident with a deepening of the upper water column layer in which dZn and dCo were depleted (a “zincocline”), with depleted surface concentrations of dZn (and dCo) observed at greater depths south of the equator (Fig. 6b, c).”
- We think that the inclusion of ZCRP spectral counts vs. dZn concentration data in both absolute scale (main text, Fig.6f,g) and in log scale (Fig. S7) is useful for the reader— the absolute scale plots in Fig.6f,g, demonstrate that many homologs were not detected at certain [dZn] (0 spectral counts, falling on the x axis), while the log scale plot in Fig. S7 allows the reader to see the increase in homologs at low dZn more easily, with the caveat that homologs with 0 detected hits at certain [dZn] are now assigned 1 hit in order to represent them on a log scale.

The following edits are also suggested in order to help clarify the text:

P.2 ll.35-36 “Among the micronutrients required for diatom metabolism, zinc (Zn) is known to be particularly important”.

- Changed as suggested

p.2 l.44 replace “implicate” with “points to”.

- We request to leave as “implicates”, as “points to” makes it sound like the data addresses mechanisms that are already characterized and understood.

p.6 l.133 replace “imply” with “mean”.

- Changed as suggested

p.6 l.134-135 “...cellular function since the C-terminal domain, which is thought to be an indicator of protein-protein interaction specificity, is quite different in these two proteins”.

- Changed as suggested

p.7 l.146 “To date, connections between COG0523 proteins and utilization of Zn and Co have been explored primarily in prokaryotic organisms”.

- Changed as suggested

p.7 ll.152-153 delete “of life”.

- Changed as suggested

p.7 ll.153-155 “Having been the first to report the presence of Zn responsive COG0523 proteins in marine diatoms, we here present additional evidence in the form of genetic, physiological, protein localization and field data.”

- We request to keep the wording the same (“We previously introduced the presence of Zn responsive COG0523 proteins in marine diatoms 24 and present genetic, physiological, characterization of localization, and field observations in this current study.”) as it flows logically.

p.8 l.161 “...organisms in which they are found”.

- Changed as suggested

p.8 l.163 replace “yet” with “whereas”.

- Changed as suggested

p. 1.166 “..., and that further metal binding and affinity assays may confirm...”

- We have changed this to “...are likely both cognate metals for diatom ZCRP-A. Further metal binding and affinity assays can confirm and characterize metal binding in this protein.”

p.9 l. 201 “...implying that their function and regulation are independent of ZCRP-B”.

- Changed as suggested

p.10 ll. 216-219 “...fits the description of a high-affinity Zn uptake system observed in marine algae that is known to be induced at low free Zn²⁺ concentrations, suggesting that...”

- Changed as suggested

p.13 ll.286-287 replace “lend further evidence to” with “provide further evidence for”.

- Changed as suggested

p.15 l.329 replace “dZn” with “Dissolved Zn”.

- We define dissolved Zn as “dZn” in the introduction and switch to using dZn throughout, we will keep as “dZn” here for consistency.

p.15 l.331 replace “dCo” with “Dissolved Co”.

- Keeping dCo for consistency as above

p.16 l.350 replace “implicates” with “implies”.

- Changed as suggested

p.16 l.355 replace “...Zn contamination. Yet...” with “...Zn contamination, yet...”

- Changed as suggested

p.16 l.356 replace “...surface ocean (e.g., Middag et al., 2019), and given...” with “...surface ocean (e.g., Middag et al., 2019). Given...”

- Changed as suggested

p.16 l.361 “...concentrations of ~1 pM...”

- Suggestion is how text is written as is

p.16 l.363 replace “..., similar to how the...” with “..., just as...”

- Changed as suggested

p.16 l.364 replace “have” with “has”.

- Changed as suggested

p.16 l.367 “...may, therefore, be...”

- Keeping text as is, “Our observations of homologous proteins among multiple eukaryotic protists throughout the South Pacific imply that Zn scarcity may be far more prevalent than previously recognized...”

p.17 1.371 “These previously uncharacterized proteins are...”

- Keeping “These findings describe...”

p.17 1.374 replace “during” with “under”.

- Changed as suggested

p.17 ll.376 substitute “with” for “but”.

- Changed as suggested

p.17 1.379 substitute “also” for “furthermore”.

- Changed as suggested

p.18 1.391 “...with constant light (40 $\mu\text{mol photon m}^{-2} \text{s}^{-1}$).

- Changed to “*Chaetoceros sp.* RS-19 cultures were maintained at 4°C under constant fluorescent lighting (40 $\mu\text{mol photon m}^{-2} \text{s}^{-1}$.” to match descriptions of other diatoms.

p.18 1.393 “*Thalassiosira pseudonana* CCMP1335...”.

- Keeping as *T. pseudonana* CCMP1335 since species name is spelled out at top of paragraph.

p.18 1.400 “techniques”.

- Keeping as “sterile technique” since “sterile technique” is itself a set of specific practices and procedures.

p.18 1.403 “Milli-Q”.

- Changed as suggested

p.18 1.408 omit “that used by”.

- Changed as suggested

p.19 1.415 substitute “in” for “within”.

- Changed as suggested

p.19 1.424 “total metal concentrations”.

- Changed as suggested

p.19 1.430 “...nitric acid containing 1 ppb indium before being analyzed...”

- Changed as suggested

p.20 ll. 441 and 443 substitute “The extract” for “Biomass”.

- Changed as suggested

p.20 ll. 448 and 449 substitute “Ten μL ” for “10 μL ”.

- Changed as suggested

P.21 ll.468-469 “...with a precursor-ion mass tolerance of 100 ppm and product-ion mass tolerance of 0.02 Da”.

- Changed as suggested

p.21 l.479 New paragraph after “...protein FDR.”?

- Created new paragraph as suggested

REVIEWERS' COMMENTS

Reviewer #2 (Remarks to the Author):

The authors have addressed my questions and queries.

Reviewer #3 (Remarks to the Author):

REVIEW: NCOMMS-21-45671A

Adaptive Responses of Marine Diatoms to Zinc Scarcity and Ecological Implications [Kellogg et al.]

The authors have done a good job in addressing the Reviewers' concerns, incorporating most of their suggestions and providing a sound rationale where these weren't implemented; for example, when retaining the predicted ZCRPB transmembrane domain plot in Fig. 4 and the log plot of ZCRP spectral counts vs. dZn concentration data in Fig. S7. I believe that changes such as the inclusion of a new supplementary Fig. S1 showing the tryptic peptides matched by LC-MS/MS data in ZCRP-A and ZCRP-B under both low Zn²⁺ and low Co²⁺ conditions, and additional references relating to high-affinity binding sites in COG0523 proteins and the involvement of putative membrane-associated Zn-binding protein ZCRP-B in cellular uptake (also noted by Reviewer 2) add further weight to the results and conclusions presented in the original manuscript. Given that it now reflects input from all three Reviewers, each of whom provided a particular set of recommendations, I consider the revised manuscript to be suitable for publication in Nature Communications and congratulate the authors on an excellent piece of work.

POINT BY POINT RESPONSE TO REVIEWERS (second revision)

“Adaptive Responses of Marine Diatoms to Zinc Scarcity and Ecological Implications”
by Riss M. Kellogg, Mark A. Moosburner, Natalie R. Cohen, Nicholas J. Hawco, Matthew R. McIlvin, Dawn M. Moran, Giacomo R. DiTullio, Adam V. Subhas, Andrew E. Allen, and Mak A. Saito.

(Author responses in blue)

REVIEWERS' COMMENTS

Reviewer #2 (Remarks to the Author):

The authors have addressed my questions and queries.

- Thank you for your feedback.

Reviewer #3 (Remarks to the Author):

REVIEW: NCOMMS-21-45671A

Adaptive Responses of Marine Diatoms to Zinc Scarcity and Ecological Implications [Kellogg et al.]

The authors have done a good job in addressing the Reviewers' concerns, incorporating most of their suggestions and providing a sound rationale where these weren't implemented; for example, when retaining the predicted ZCRPB transmembrane domain plot in Fig. 4 and the log plot of ZCRP spectral counts vs. dZn concentration data in Fig. S7. I believe that changes such as the inclusion of a new supplementary Fig. S1 showing the tryptic peptides matched by LC-MS/MS data in ZCRP-A and ZCRP-B under both low Zn²⁺ and low Co²⁺ conditions, and additional references relating to high-affinity binding sites in COG0523 proteins and the involvement of putative membrane-associated Zn-binding protein ZCRP-B in cellular uptake (also noted by Reviewer 2) add further weight to the results and conclusions presented in the original manuscript. Given that it now reflects input from all three Reviewers, each of whom provided a particular set of recommendations, I consider the revised manuscript to be suitable for publication in Nature Communications and congratulate the authors on an excellent piece of work.

- Thank you for your feedback.